# Developmental beta-cell death orchestrates the islet's inflammatory milieu by regulating immune system crosstalk

Mohammad Nadeem Akhtar[1,2,3], Alisa Hnatiuk[1,2,3], Luis Delgadillo-Silva [ID] [4], Shirin Geravandi[5], Katrin Sameith[6], Susanne Reinhardt[6], Katja Bernhardt[3], Sumeet Pal Singh[7], Kathrin Maedler [ID] [5], Lutz Brusch [ID] [8] & Nikolay Ninov [ID] [1,2] [✉]

## Abstract

**While pancreatic beta-cell proliferation has been extensively studied, the role of cell death during islet development remains incompletely understood. Using a genetic model of caspase inhibition in beta cells coupled with mathematical modeling, we here discover an onset of beta-cell death in juvenile zebrafish, which regulates beta-cell mass. Histologically, this beta-cell death is underestimated due to phagocytosis by resident macrophages. To investigate beta-cell apoptosis at the molecular level, we implement a conditional model of beta-cell death linked to Ca²⁺ overload. Transcriptomic analysis reveals that metabolically-stressed beta cells follow paths to either de-differentiation or apoptosis. Beta cells destined to die activate inflammatory and immuno-regulatory pathways, suggesting that cell death regulates the crosstalk with immune cells. Consistently, inhibiting beta-cell death during development reduces pro-inflammatory resident macrophages and expands T-regulatory cells, the deficiency of which causes premature activation of NF-kB signaling in beta cells. Thus, developmental cell death not only shapes beta-cell mass but it also influences the islet's inflammatory milieu by shifting the immune-cell population towards pro-inflammatory.**

**Keywords** Type 1 Diabetes; Excitotoxicity; Dedifferentiationp; T Regulatory Cell; Macrophage
**Subject Categories** Development; Immunology; Metabolism

## Introduction

Similar to any other cell type in the body, beta-cell turnover during development is orchestrated by a balanced rate of proliferation and apoptosis. In contrast to proliferation, the relative contribution of apoptosis towards beta-cell turnover has not been well documented (Meier et al, 2005), and it remains unknown how beta-cell death shapes the populations of resident immune system cells. This question is of fundamental importance in the context of Type 1 diabetes (T1D).

T1D is a progressive autoimmune disease in which auto-reactive immune cells selectively destroy the beta cells (Kawasaki, 2014). It is also known as juvenile diabetes because of the early age of onset. Strikingly, the appearance of autoantibodies (AAbs) against islet antigens (seroconversion), which reflects the start of autoimmunity, often occurs around the 12th months of age in children, suggesting a connection between early-life pancreas remodeling and the interactions between beta cells and the immune system (Parikka et al, 2012; Ziegler and Bonifacio, 2012). In support of this idea, Warncke et al, recently identified defects in glucose handling in children with a genetic risk of Type 1 diabetes. These defects precede by several months the onset of seroconversion (Warncke et al, 2022). This knowledge suggests that cellular and metabolic changes in beta cells during early-life islet remodeling may impinge on the interactions between the islet and the immune system, promoting the misguided presentation of beta-cell antigens to T and B-cells. However, the nature of the events that take place in the islets during juvenile pancreas maturation and their role in immune cell surveillance remains to be investigated. An intriguing candidate to take into account is the occurrence of cell death that may accompany pancreas maturation, however, whether cell death really takes place and whether it has pro- or anti-inflammatory features is a topic of debate with conflicting results.

[1]Centre for Regenerative Therapies TU Dresden, Dresden 01307, Germany. [2]Paul Langerhans Institute Dresden of the Helmholtz Center Munich at the University Hospital Carl Gustav Carus of TU Dresden, German Center for Diabetes Research (DZD e.V.), Dresden 01307, Germany. [3]Technische Universität Dresden, CRTD, Center for Molecular and Cellular Bioengineering (CMCB), Fetscherstraße 105, 01307 Dresden, Germany. [4]Cardiometabolic Axis, CR-CHUM and University of Montreal, Montreal, QC, Canada. [5]Centre for Biomolecular Interactions Bremen, University of Bremen, 28359 Bremen, Germany. [6]DRESDEN-concept Genome Center, DFG NGS Competence Center, c/o Center for Molecular and Cellular Bioengineering (CMCB), Technische Universität Dresden, 01307 Dresden, Germany. [7]IRIBHM, Université Libre de Bruxelles (ULB), 1070 Brussels, Belgium. [8]Centre for Interdisciplinary Digital Sciences (CIDS), Information Services and High Performance Computing (ZIH), Technische Universität Dresden, 01187 Dresden, Germany. [✉]E-mail: nikolay.ninov@tu-dresden.de

Previously, a study using a mathematical model of beta-cell turnover in the developing rat pancreas proposed the existence of a wave of neonatal beta-cell apoptosis, which subsequently declines in adults (Finegood et al, 1995). Using histological assessment of the rat pancreatic islet, it was shown that the general apoptotic index of islet cells peaks around 13–17 days after birth and then decreases dramatically during adulthood (Scaglia et al, 1997; Petrik et al, 1998). Moreover, Scaglia et al, used propidium iodide to identify apoptotic nuclei based on their condensed morphological appearances and also proposed that pancreas remodeling is associated with cell death (Scaglia et al, 1997). The actual occurrence of beta-cell death may be masked by the rapid removal of dying cells prior to the morphological changes associated with cell death, making it difficult to define the significance of this process for beta-cell mass control. Moreover, the consequences of physiological cell death on the inflammatory milieu of the islet remain unknown.

To address these questions, we use an array of tools, which include genetic engineering, mathematical modeling, and single-cell transcriptomics, to study beta-cell death and its role in regulating the islet's immune component. We show that metabolically stressed beta cells follow paths toward either dedifferentiation or cell death and that this process is associated with the expression of genes involved in immune cell crosstalk. We further define a developmental stage during which an onset of beta-cell death takes place in the islet. By blocking this cell death genetically, we reveal that the developmental beta-cell death orchestrates the islet's inflammatory milieu by regulating the crosstalk with immune cells, including macrophages and T regulatory cells, the latter playing an active anti-inflammatory role. We also find an increase in macrophage recruitment to the islets in donors with T1D.

# Results

## Genetic inhibition of caspase leads to beta-cell expansion in juvenile zebrafish

Historically, it has been very challenging to study cell death in vivo due to difficulties in its histological detection (Dixon and Lee, 2023; Surh and Sprent, 1994). To overcome this hurdle and be able to define the contribution of beta-cell death to islet turnover, we generated a transgenic zebrafish *Tg(ins:p35)*, which expresses the caspase inhibitor p35 under the insulin promoter (Fig. 1A,B). p35 is a baculovirus pan-caspase inhibitor that has been shown to block developmental cell death in *Drosophila* and mouse (Hay et al, 1994; Clavería et al, 2013). We crossed the *Tg(ins:p35)* line to the *Tg(ins:mCherry)* reporter line, which expresses a bright red fluorescent protein in the beta cells. This enabled us to quantify the beta cell number during development. We imaged the primary islets from *Tg(ins:mCherry)* and the double transgenic *Tg(ins:p35); Tg(ins:mCherry)* zebrafish at 5 dpf (days post fertilization), 15 dpf, and 30 dpf (Fig. 1C). At the early developmental stages i.e., at 5 and 15 dpf, there was no significant difference in the number of beta cells between the controls and p35-expressing animals. However, at 30 dpf, we observed a 1.5-fold increase in the number of beta cells when compared to clutch mate controls (Fig. 1E). Analysis of intermediate developmental stages showed that the excess beta cells first appear by 20 dpf in *Tg(ins:p35)* (Fig. 1E; Appendix Fig. S1B).

We wondered if the increase in beta-cell number in *Tg(ins:p35)* animals was due to increased proliferation. To this end, we performed PCNA (proliferative cell nuclear antigen) staining of the pancreatic islet of WT (wild-type) and *Tg(ins:p35)* animals at different stages of development i.e., 15, 25, and 30 dpf (Fig. 1D). However, we did not observe any significant difference in the percentage of proliferating beta cells between the WT and *Tg(ins:p35)* animals (Fig. 1F). EDU (5-ethynyl 2′-deoxyuridine) incorporation analysis also confirmed these results (Appendix Fig. S1C,D).

We also questioned if genetic overexpression of p35 alters the functionality of beta cells. We performed live imaging of zebrafish larvae in order to assess glucose-stimulated calcium influx of pancreatic beta cells using a beta-cell specific *Tg(ins:GCaMP6s)* reporter line (Singh et al, 2017). Stimulation with 12.5 mM glucose lead to a synchronized beta-cell response in both *Tg(ins:GCaMP6s)* and *Tg(ins:p35); Tg(ins:GCaMP6s)* double transgenic larvae, as observed by the increase in GCaMP fluorescence intensity (Appendix Fig. S2A,B). Furthermore, we measured glucose in WT and *Tg(ins:p35)* at 5 dpf and 30 dpf and did not find differences (Fig. 2C,D). We also found no significant differences in the blood glucose levels of 6-month-old WT and *Tg(ins:p35)* animals (Appendix Fig. S2E) while the cell mass remained increased (Appendix Fig. S2F). This suggests that genetic caspase inhibition in juvenile zebrafish leads to beta-cell expansion without perturbing beta-cell proliferation and function.

## A mathematical model estimates that apoptosis participates in the beta-cell turnover process

As shown by Finegood et al, the rate of beta-cell death can be calculated from a simple mathematical model of beta-cell turnover incorporating measurements of the temporal change of total beta-cell mass together with beta-cell neogenesis and proliferation rates (Finegood et al, 1995). However, they had considered all beta cells equally proliferative for the rat pancreas, whereas previous data in zebrafish show that a minor fraction of beta cells is able to proliferate (Singh et al, 2017; Hesselson et al, 2009) (Fig. 1F). We, therefore, had to extend the simple model by considering two subpopulations, one of the proliferative beta cells with time-dependent number $N_p(t)$ and another of quiescent beta cells with time-dependent number $N_q(t)$, respectively, as reported previously (Singh et al, 2017). Figure 2A shows a sketch of the model with all considered processes and the derived differential equations are shown in the methods section on mathematical modeling. The model predicts $Np(t)$ and $Nq(t)$, and then the sum $N(t) = Np(t) + Nq(t)$ is calculated and compared to the experimental cell number counts. While Finegood et al, had inserted data from a single experimental condition, namely WT rat pancreases, into their model, we incorporated data from two experimental conditions: WT and *Tg(ins:p35)* zebrafish. We estimate the six free model parameters (see Methods section Mathematical Modeling) by fitting the simulated $N(t)$ to 66 individual data points from the two combined experimental conditions (see Fig. 1E; Appendix Fig. S1B), enforcing the same values for all parameters for both conditions except for the death rate, which is a free parameter for the WT and set to zero for the p35 genotype, respectively. The best-fitting model result is shown in Fig. 2A' and the estimated parameter values are given in the methods section on mathematical

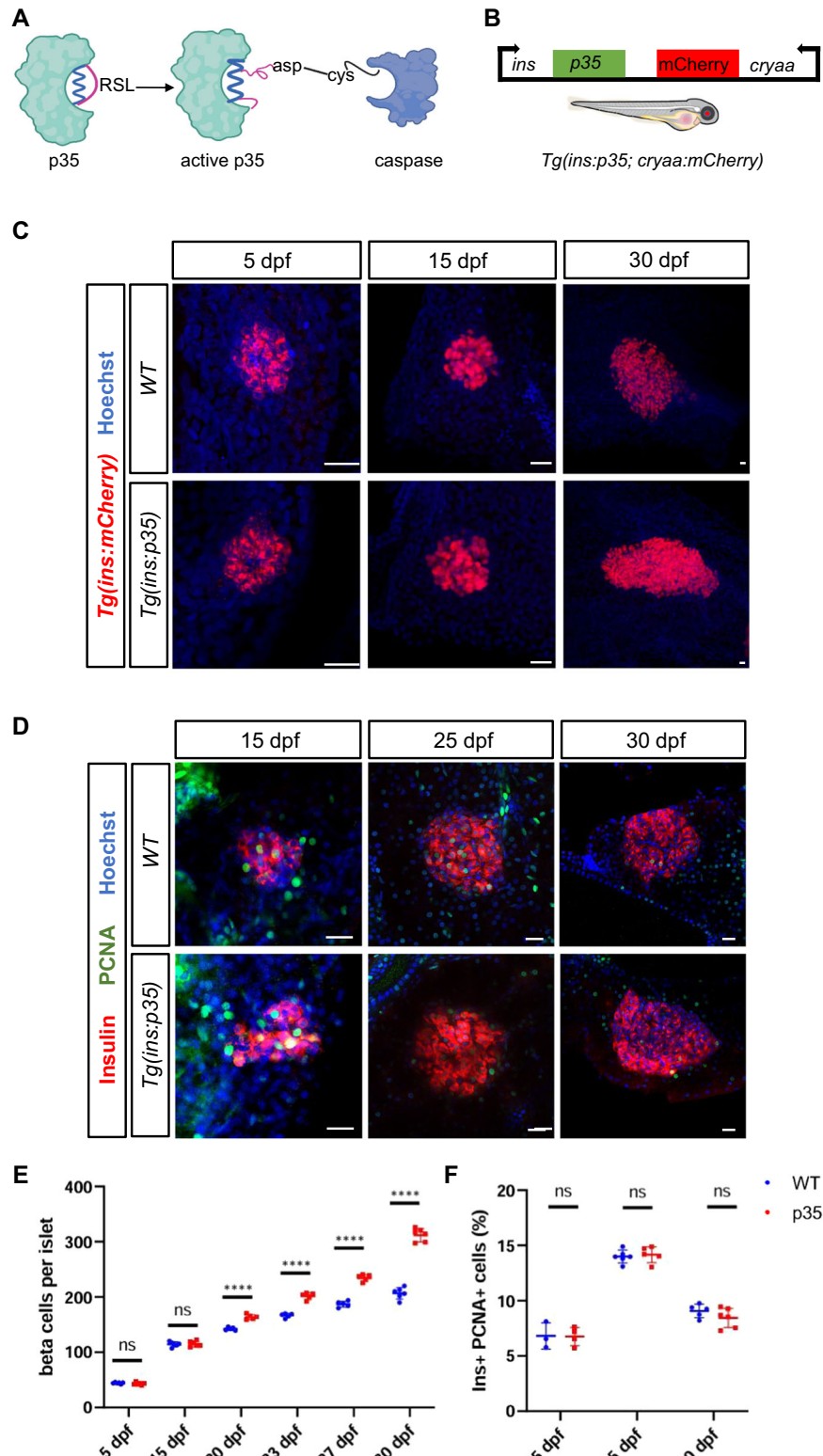

modeling. Importantly, a beta-cell death rate of $3.0 \pm 0.6\%$ per day is inferred by the model fit which starts from $16.3 \pm 1.6$ dpf and continues to at least 30 dpf, corresponding to 4 or 5 dying cells throughout the entire islet per day at 20 dpf and 9 dying cells at 30 dpf. Interestingly, this model with continuous cell death from an onset time of $16.3 \pm 1.6$ dpf onwards performed better than a variant

with a one-day burst of apoptosis at an optimized time of around 20 dpf (Fig. 2B). Moreover, to approach the role of cell death from the opposite starting point, we have considered a model variant in which cell death was assumed to be absent and its rate therefore set to zero for the WT condition (Fig. 2C). Then all other model parameters were estimated by again fitting the data points from two

**Figure 1.  Genetic overexpression of *p35* leads to beta-cell expansion in juvenile zebrafish.**

(A) Schematic showing the baculovirus P35 protein and its mechanism of caspase inhibition. RSL indicates the reactive site loop of P35 (B) Diagrammatic representation showing the genetic construct used to generate the transgenic model of caspase inhibition in beta cells. *p35* was cloned under the control zebrafish insulin promoter. mCherry expression under the control of the crystalline (*cryaa*) promoter serves as a marker of transgenic animals. (C) Representative confocal images (maximum projection) of primary islets from *Tg(ins:mCherry)* and *Tg(ins:p35); Tg(ins:mCherry)* at 5, 15, and 30 dpf. Immunostaining against insulin is used to mark the beta cells (in red) while the nuclei are stained with Hoechst (in blue) of 5 dpf WT and *Tg(ins:p35) larvae*. The beta cells of 15 and 30 dpf WT and *Tg(ins:p35)* animals are marked using *Tg(ins:mCherry)* reporter in red. Scale bar, 20 μm. (D) Representative confocal images (single plane) of primary islets from WT and clutch mate *Tg(ins:p35)* animals at 15, 25, and 30 dpf. Immunostaining against insulin (red) and PCNA (green) marks the beta cells and the proliferating cells, respectively. The nuclei are stained using Hoechst. Scale bar, 20 μm. (E) Quantifications of the number of beta cells in *Tg(ins:mCherry)* and *Tg(ins:p35); Tg(ins:mCherry)* animals at 5, 15, 20, 23, 27, and 30 dpf. Each dot represents the number of beta cells per islet. Error bars are mean ± SD. Unpaired two-tailed *t*-test with Welch's correction, 5 dpf (ns not significant, $p = 0.141307$), 15 dpf (ns not significant, $p = 0.903301$), 20 dpf (****$p = 0.000025$), 23 dpf (****$p = 0.000006$), 27 dpf (****$p = 0.000001$), and 30 dpf (****$p = 0.0000000194$). $n =$ at least 5 independent samples per group. (F) Quantification showing the percentage of insulin and PCNA double-positive cells per islet in WT and *Tg(ins:p35)* animals at 15, 25, and 30 dpf. $n =$ at least 3 independent samples per group at each developmental stage. Error bars are mean ± SD. Unpaired two-tailed *t*-test with Welch's correction, ns not significant. Source data are available online for this figure.

experimental conditions. The result is shown in Fig. 2C' and shows large discrepancies for both conditions at 30 dpf, thereby falsifying the assumption of the absence of cell death. We have confirmed that the identified mechanism yields the same results in a deterministic model (Fig. 2A–C) and in a stochastic cell-based model (Fig. 2D and Movie EV1), implemented in the software Morpheus (Starruß et al, 2014) with parameter estimator FitMultiCell (Alamoudi et al, 2023).

In order to corroborate the estimate of cell death provided by the mathematical model, we performed a histological assessment of the pancreatic islet at different stages of pancreas development, including 15 and 30 dpf. We performed TUNEL (terminal deoxynucleotidyl transferase dUTP nick end labeling) assay using in *Tg(ins:mCherry)* transgenic line. Using this approach, we detected traces of beta-cell apoptosis during pancreas development at 30 dpf, however, the detected rate did not match the rate that was predicted by the mathematical model of beta-cell turnover (Appendix Fig. S3). The primary reason for this discrepancy could be that apoptotic cells are rapidly removed by professional phagocytes such as macrophages making it difficult to histologically estimate the true extent of beta-cell apoptosis.

## Macrophage islet colonization correlates with the onset of beta-cell apoptosis during pancreas development

Our mathematical model of islet turnover in the p35 model provides hints that developmental beta-cell apoptosis starts slightly before 20 dpf, yet, we failed to detect extensive signs of this process (Appendix Fig. S3). This can be due to the rapid removal of beta-cell corpses by professional phagocytes. To test this hypothesis, we first carefully assessed the timing of macrophage colonization of the islet by performing time course assessment of the pancreatic macrophages using *Tg(ins:YFP); Tg(mpeg:mCherry)* double transgenic fish. At earlier stages of development i.e., at 5 and 15 dpf, we did not observe any evidence of macrophages infiltrating the islet, consistent with our model that beta-cell death is negligible. However, by 21 dpf, we observed a significant increase in the number of macrophages present in and around the islet, which persisted until later stages of development i.e., 42 dpf (Fig. 3A,B).

If the phagocytic clearance of apoptotic cells hinders the histological detection of cell death, we expect that the reduction in the number of macrophages would result in an increased accumulation of beta-cell corpses in the islet during pancreas

development. In order to test this hypothesis, we took advantage of the *irf8* (*interferon regulatory factor 8*) mutant line. *IRF8* is known to play an important role in macrophage development and innate immunity. Genetic inhibition of *irf8* leads to a significant reduction in the number of macrophages in zebrafish (Shiau et al, 2015). We analyzed the *irf8* mutants in the background of a beta-cell specific reporter line *Tg(ins:YFP)*. We performed a TUNEL assay in the islet of 30 dpf *Tg(ins:YFP);irf8$^{+/+}$* and *Tg(ins:YFP);irf8$^{-/-}$* animals. As hypothesized, we observed an increase in TUNEL-positive beta cells in the islet of *Tg(ins:YFP);irf8$^{-/-}$* animals (Fig. 3C,D). Besides being TUNEL-positive, there were additional morphological appearances associated with apoptosis, such as shrinkage of cytoplasm and rounding up of nuclei (Fig. 3C'). We did not observe TUNEL-positive beta cells in the islet of *Tg(ins:YFP);irf8$^{+/+}$* animals, highlighting the critical role of macrophages in the clearance of apoptotic cells during zebrafish islet development. Altogether, the stage-dependent accumulation of islet macrophages and the appearance of beta-cell corpses upon macrophage depletion argue in support of our model that the onset of beta-cell death accompanies early juvenile pancreas remodeling and is associated with macrophage recruitment.

To examine the association between beta cells and macrophages in a mammalian setting, we reanalyzed histological data previously obtained from human donors (Geravandi et al, 2021). We found an increase in macrophages within islets of organ donors diagnosed with T1D, compared to controls (Appendix Fig. S4). As shown previously, these individuals also present a reduction in beta-cell mass (Geravandi et al, 2021). It is therefore possible that similarly to zebrafish, human macrophages recruit to the islets in the course of beta-cell death.

## Chemogenic induction of Ca²⁺ excitotoxicity and ablation of TRPV-expressing beta cells

Since developmental beta-cell death is stochastic and difficult to detect, to be able to explore the molecular underpinning of cell death, we necessitated a model that would allow one to induce beta-cell death in a controlled manner in order to study the dynamics of the process. In contrast to the established nitroreductase (NTR) method for beta-cell ablation, which crosslinks the DNA and triggers complete and unrepairable damage to the cells (Edwards, 1993), we focused on creating a new model of cell exhaustion via excitotoxicity, an important component of metabolic stress in

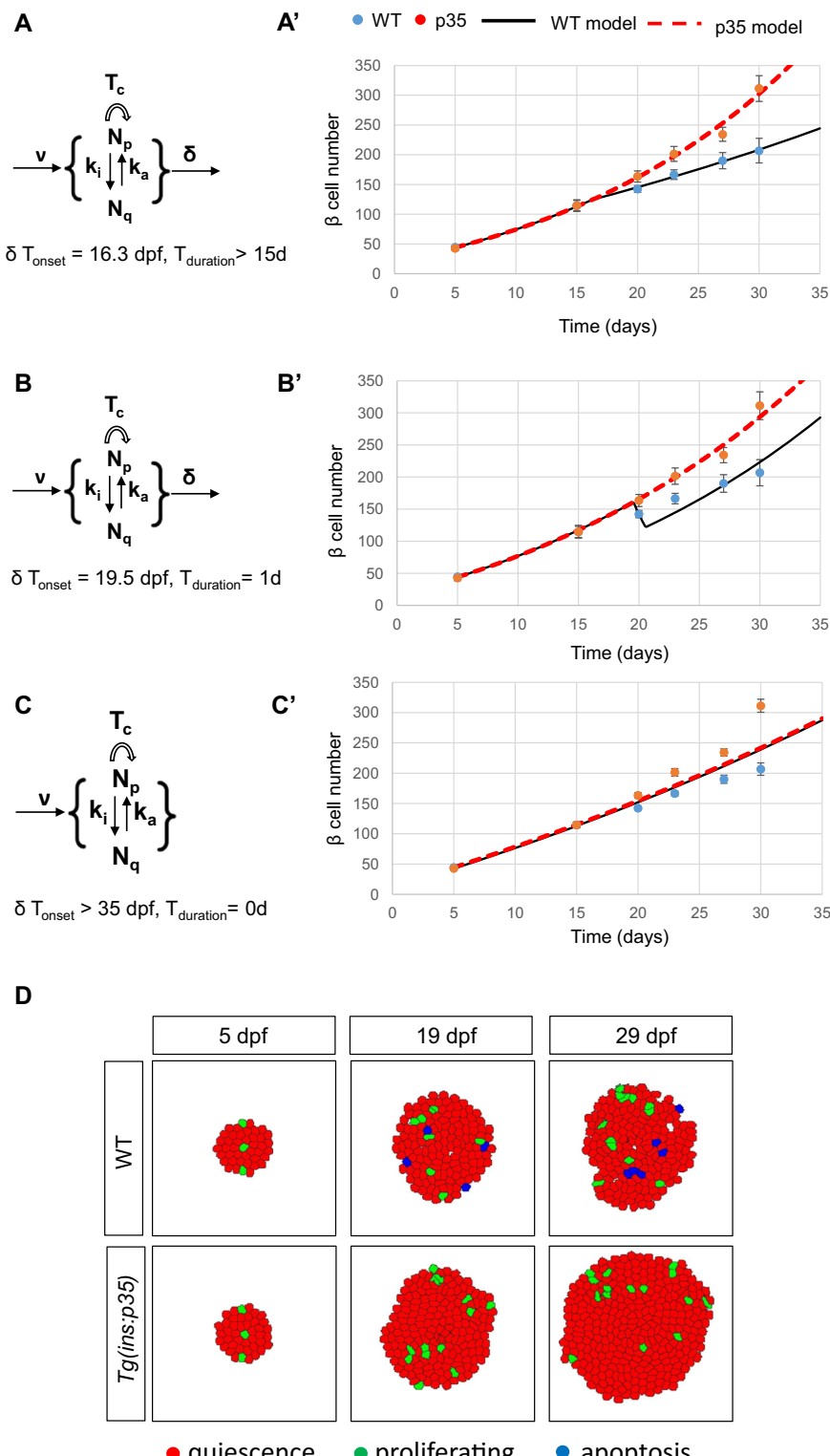

**A** δ $T_{onset}$ = 16.3 dpf, $T_{duration}$ > 15d

**B** δ $T_{onset}$ = 19.5 dpf, $T_{duration}$ = 1d

**C** δ $T_{onset}$ > 35 dpf, $T_{duration}$ = 0d

**A'**, **B'**, **C'**: ● WT ● p35 — WT model --- p35 model; β cell number vs Time (days)

**D** 5 dpf, 19 dpf, 29 dpf; WT, *Tg(ins:p35)*

● quiescence ● proliferating ● apoptosis

beta cells (Stancill et al, 2017). To this end, we generated a chemogenetic model of beta-cell excitotoxicity using the TRPV (transient receptor potential channel vanilloid) system (Caterina et al, 1997). TRPV is a six-transmembrane cation channel that can be activated by a small molecule, capsaicin (csn) (Fig. 4A).

Stimulation with csn leads to the activation of the TRPV channel causing an influx of calcium ion. We took advantage of the rat TRPV channel as the zebrafish TRPV ortholog is insensitive to csn (Gau et al, 2013). We created a transgenic zebrafish line, *Tg(ins:TRPV1, cryaa:cerulean)* that expresses TRPV under the

**Figure 2. Mathematical model.**

(A) Sketch of five processes and two subpopulations of beta cells considered in the mathematical model. $N_p(t)$ is the time-dependent number of proliferative beta cells and $N_q(t)$ of quiescent beta cells. Proliferative cells from $N_p(t)$ may reversibly switch to the quiescent state $N_q(t)$ with a rate $k_i$ and back to proliferation with a rate $k_a$. Cell cycle duration is $T_c$, neogenesis rate is $v$, and death rate is $\delta$. The latter two processes affect both subpopulations. Cell death is initially absent and starts at an estimated onset time of 16.3 dpf. (A') Results of the fitted model for WT (black) and p35 (red) conditions, respectively. Experimental data at the time points 5, 15, 20, 23, 27, and 30 dpf (from Fig. 1E; Appendix Fig. S1B) is shown as the mean (circle) and 2 SD (risers). Color code as shown on the top of panel A': blue and red circles represent experimental data of beta-cell numbers in WT and p35, respectively; black solid line and red dashed line depict the model simulation of WT and p35. $n$ = at least 5 independent samples per group at each developmental time point. (B-B') Results for the model variant with 1-day short burst of cell death around 20 dpf. This variant is not able to represent the shallower slope for WT cell count data after 20 dpf. (C-C') Results for model variant without cell death for both experimental conditions. All other model parameters were estimated; however, the best-fitting curves show large discrepancies for both conditions at 30 dpf, thereby falsifying the assumption of absent cell death. Error bars in A'-C' are SEM. (D) Snapshots from simulation movie (Movie EV1) of a stochastic, cell-based model with the same mechanisms and parameter values as in (A) confirms the results from (A). Source data are available online for this figure.

regulatory elements of the insulin gene (Fig. 4B), as it has been previously done for neurons (Chen et al, 2016). To assess whether *Tg(ins:TRPV)* animals exhibited Ca²⁺ influx following csn treatment, we analyzed the beta-cell calcium response using the transgenic reporter line *Tg(ins:GCaMP6s)* in larvae. While, there was no alteration in the calcium activity of beta cells in the control larvae, *Tg(ins:GCaMP6s)*, there was a robust increase in Ca²⁺ influx in the beta cells of *Tg(ins:TRPV); Tg(ins:GCaMP6s)* larvae after csn treatment (Fig. 4C,D; Movies EV1, 2).

In order to quantify the beta-cell number post csn treatment, we crossed the *Tg(ins:TRPV)* line with a transgenic line that expresses the brightly fluorescent protein Kaede under the control of the *insulin* promoter, *Tg(ins:Kaede)*. We treated *Tg(ins:Kaede)* and the double transgenic larvae *Tg(ins:TRPV); Tg(ins:Kaede)* larvae with 40 μM csn for 48 h i.e., from 3 to 5 dpf. We observed that there was a significant reduction in the number of beta cells of *Tg(ins:TRPV); Tg(ins:Kaede)* larvae as compared to control larvae, *Tg(ins:Kaede)* (Fig. 4E,F). Furthermore, TRPV transgenic larvae were hyperglycemic when compared to WT siblings post csn treatment (Fig. 4G). Despite the loss of cells, we failed to detect the appearance of beta-cell corpses in this model, akin to the situation we observed during normal development. However, by genetically ablating the macrophages using *Tg(mpeg1:EYFP-NTR)* transgenic line, which harbors a DNA construct containing the bacterial NTR enzyme driven by the mpeg promoter we observed incidence of beta-cell debris characterized by cytoplasmic shrinkage and nuclear condensation accumulating in the islet of macrophages depleted animals (Appendix Fig. S5A,B). Therefore, these results corroborate our hypothesis that the corpses of dying beta cells in the TRPV model of excitotoxicity are rapidly removed by macrophages. This leads to the underrepresentation of the actual rate of cell death observed in histological analysis in vivo.

To determine whether the decrease in the number of beta cells is due to the ablation of beta cells and not due to reduced proliferation or formation of new beta cells, we followed the fate of beta cells during csn treatment via photoconversion. Using *Tg(ins:TRPV);Tg(ins:Kaede)*, we photoconverted the fluorescence of the Kaede protein from green to red by exposing the larvae to UV light at 3 dpf. Thereafter, we treated the larvae with csn and EdU for 48 h. The latter was used to quantify the proliferating beta cells. Using this strategy, beta cells that survive the ablation will co-express red and green fluorescent Kaede (indicated by yellow overlap), the newly formed beta cells will express only green fluorescent Kaede, while the proliferating cells will be marked with EdU incorporation. We quantified the percentage of proliferating

beta cells i.e., the number of beta cells that incorporated EdU over the total number of beta cells in the *Tg(ins:Kaede)* and *Tg(ins:TRPV); Tg(ins:Kaede)* larvae. We did not find any significant difference in the percentage of proliferating beta cells between *Tg(ins:Kaede)* and *Tg(ins:TRPV); Tg(ins:Kaede)* larvae (Fig. 4H,I). In addition, we quantified the green-only beta cells to check if differentiation is perturbed upon chronic excitotoxicity. We did not find any significant difference in the number of green-only beta cells between *Tg(ins:Kaede)* and *Tg(ins:TRPV);Tg(ins:Kaede)* group. However, there was a clear reduction in the number of pre-existing beta-cells (red and green-positive cells) after csn-treatment (Appendix Fig. S6).

## Single-cell transcriptomics reveals heterogeneity among beta cells in response to Ca²⁺ excitotoxicity

The model of controllable induction of excitotoxicity provided us with a platform to determine the transcriptional changes associated with beta-cell death. To this end, we performed single-cell RNA sequencing on beta cells isolated from *Tg(ins:TRPV); Tg(ins:Kaede)* larvae treated with csn for 24 or 48 h starting at 3 dpf. Beta cells isolated from *Tg(ins:Kaede)* larvae treated with csn were used as control (Fig. 5A). Post treatment, the pancreas of the larvae was manually dissected and then enzymatically dissociated. After dissociation, the GFP-positive viable (calcein stained) beta cells were FACS (Fluorescence Activated Cell Sorting) sorted and profiled using Smartseq2 pipeline (Appendix Fig. S7A). The transcriptome data was subjected to quality control that included genes detected per cell and the percentage of mitochondrial reads detected per cell. This yielded a total of 650 beta cells (171 for controls; 150 cells for the 24-h treatment group; and 329 for the 48-h treatment group) for downstream analysis. Assessment of the top ten enriched genes showed that *ins* is the highest expressed gene in all three groups, which further confirmed that isolated GFP-positive cells were beta cells (Appendix Fig. SB). Beta cells from all three treatment groups were visualized using Uniform Manifold Approximation and Projection (UMAP), an algorithm that unbiasedly grouped the cells into five different clusters based on the similarity of gene expression (Fig. 5B). One notable feature is that beta cells from the 48-h csn-treatment group are very heterogeneously distributed, i.e., they are found across all the five clusters albeit in different proportions, indicating that stressed beta cells take on different fates. Interestingly, we found that cluster 4 largely comprised of beta cells from the 48-h csn-treatment group but no control cells and only a small contribution from beta cells

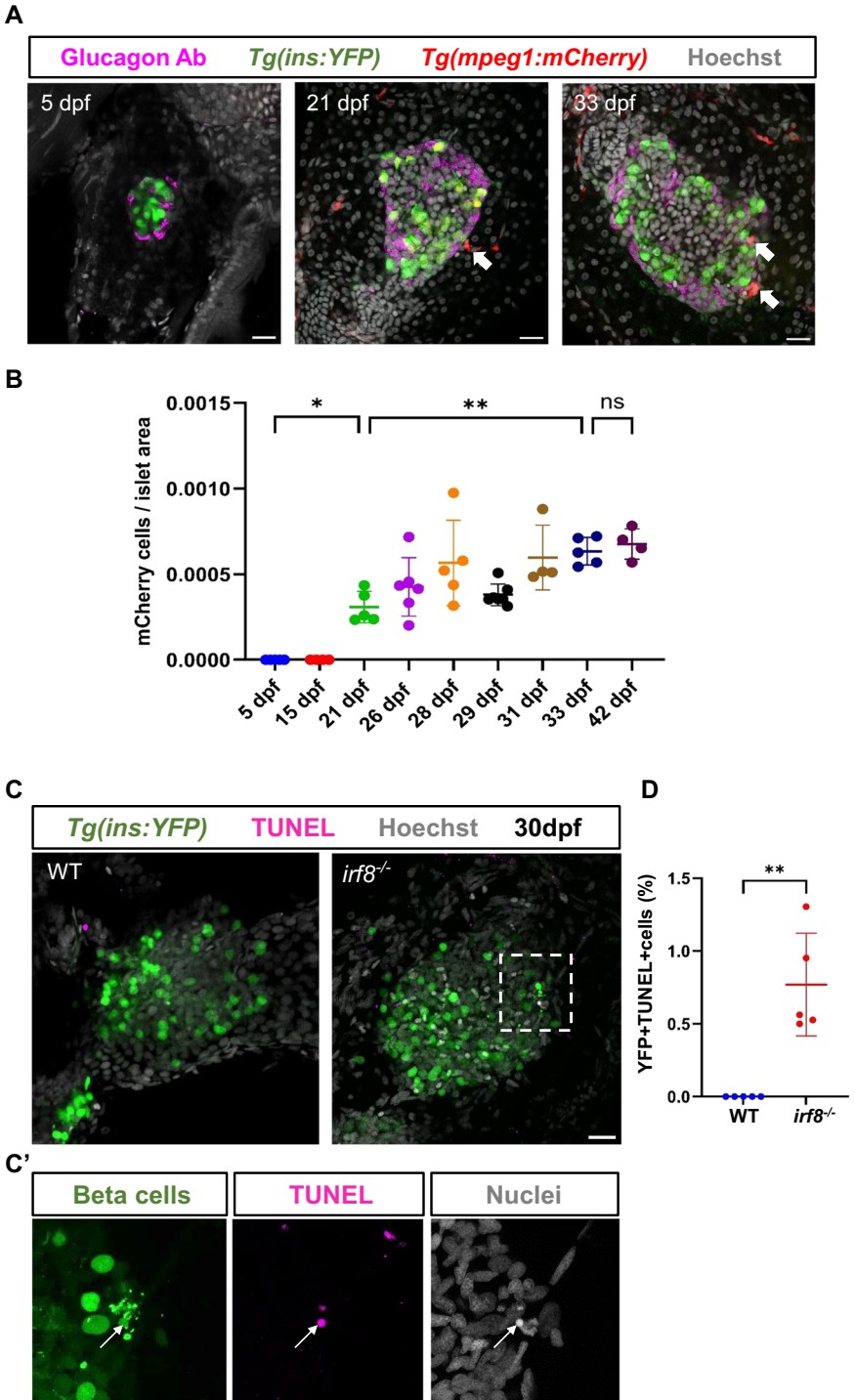

from the 24-h csn-treatment group. At the same time, cluster 5 contained exclusively beta cells from the 48-h csn-treatment group (Fig. 5C). Closer examination revealed that genes related to cell death, such as *casp-3* (annotated as CR925773.1), *acin1b*, and *bida* (Luo et al, 1998) were found to be enriched in beta cells from cluster 4 (Fig. 5D). Furthermore, *nfkbiaa* and *jdp2b* which are known markers of inflammation (Janjuha et al, 2018; Yang et al, 2020) were upregulated. Besides cell death and inflammation, genes

related to innate immune system activation, such as the complement complex *c3a.3* and *c3b.1* were enriched in cluster 4 (Houseright et al, 2020) (Fig. 5D). Additionally, we found that genes related to T-cell activation and proliferation such as *zap70*, *cd28b* (Salojin et al, 1999), and *ikzf2* (Xie et al, 2021) were upregulated. This suggests that beta cells in cluster 4 are showing the signature associated with inflammatory cell death and actively participating in the recruitment and activation of the immune

**Figure 3. Macrophage colonization coincides with the onset of developmental beta-cell apoptosis, and their depletion reveals developmental beta-cell apoptosis.**

(A) Confocal images (single plane) of primary islets from *Tg(ins:YFP); Tg(mpeg:mCherry)* animals at 5, 21, and 33 dpf. *Tg(ins:YFP)* marks the beta cells (green), *Tg(mpeg:mCherry)* marks the macrophages (red), while immunostaining against glucagon labels the alpha cells (in magenta) present on the periphery of the islet. Macrophages (white arrows) are observed in contact with the islet at 21 dpf while they are absent at 5 dpf. At 33 dpf, macrophages have colonized the islet. Scale bar, 20 μm. (B) Quantification showing the number of mCherry-positive cells per unit area in the pancreatic islet at different stages of development. Error bars are mean ± SD from n = at least 4 independent samples per developmental time point. One-way ANOVA with Tukey's multiple comparison test, *$p = 0.0158$, **$p = 0.0088$, ns not significant. (C) Confocal images (maximum projection) of primary islet from 30 dpf *Tg(ins:YFP);irf8+/+* and *Tg(ins:YFP);irf8−/−* animals. *Tg(ins:YFP)* marks the beta cells (green), TUNEL staining labels the apoptotic cells (magenta), while Hoechst staining labels the nuclei (gray). White arrow points to TUNEL-positive beta cells. Scale bar, 20 μm. (C') Insets show a high-magnification single plane with separate channels from the confocal stacks (corresponding to the area marked using a white dotted line in (C). Scale bar, 20 μm. (D) Quantification showing the percentage of YFP and TUNEL double-positive cells in the islet of *Tg(ins:YFP);irf8+/+* and *Tg(ins:YFP);irf8−/−* animals. Error bars are mean ± SD from *n* = 5 independent samples per group. Unpaired two-tailed *t*-test with Welch's correction, **$p = 0.0081$. Source data are available online for this figure.

system. In contrast to cluster 4, beta cells in cluster 5 did not show the upregulation of cell death and immune-related genes. Instead, they showed a more prominent downregulation of bona-fide beta-cell markers including *ins* and *pdx1*, and upregulated expression of the canonical marker of mammalian beta-cell dedifferentiation *aldh1a3* (Fig. 5D) (Kim-Muller et al, 2016; Cinti et al, 2016). In addition, several mitochondrial genes involved in oxidative phosphorylation such as *coq5* and *ndufa4* were downregulated (Dataset EV1, Dataset EV2). This indicates that beta cells in cluster 5 possibly evade the stress by dedifferentiation and reducing their identity. Furthermore, slingshot, which is a pseudotime trajectory inference analysis tool (Steer et al, 2018), showed that beta-cell clusters 2, 1, 3, 4, and 5 track along a linear continuum (Fig. 5E). This implies that under chronic Ca²⁺ elevation, a subset of stressed beta cells (cluster 5) is able to bypass cell death by undergoing dedifferentiation as a protective mechanism. Moreover, a different group of beta cells (cluster 4) succumbs to the stress and activates genes involved in inflammation and the crosstalk with the immune system (Fig. 5F).

To validate these results at the protein level, we took advantage of the *Tg(NF-kB:EGFP)* reporter line. Here, GFP is expressed upon the nuclear translocation and binding of the NF-kB dimer to the NF-kB binding site (Kanther et al, 2011). We detected activation of GFP expression in 10% of beta cells after 48 h of csn treatment (Appendix Fig. S8A–C), which is in line with the proportion of cells activating inflammation markers in our RNA-seq data (cluster 4, 12%). Interestingly, neighboring insulin-negative cells appeared to also activate NF-kB (Appendix Fig. S8B). Next, we performed Pdx1 antibody staining after csn-treatment. We observed an increased proportion of Pdx1-low beta cells compared to controls (Appendix Fig. S8D–F). These cells tend to exhibit low insulin expression, consistent with the reduced *ins* and *pdx1* expression after 48 h of csn treatment. Subsequently, using the Kaede-photoconversion approach to mark old and new beta cells, we found that pre-existing beta cells downregulate Pdx1 after TRPV activation (Appendix Fig. S9A,B). We also ablated the macrophages in order to reveal beta-cell corpses and observe the levels of Pdx1. The dying cells, which showed condensed and pycnotic nuclei, had lost their nuclear Pdx1 expression (Appendix Fig. S9C, *n* = 13 cells from *n* = 6 larvae). Forcing p35-expression in the TRPV model partially rescued the decline in beta-cell number (Appendix Fig. S9D–F), but it was unable to prevent Pdx1-downregulation (Appendix Fig. S9B). This suggests that caspase inhibition rescues the cells from apoptosis but not from dedifferentiation. Taken together with our transcriptomic

analysis, these results reveal that metabolically stressed beta cells follow paths toward apoptosis or dedifferentiation (Fig. 5F).

## Developmental beta-cell death enables the recruitment of immune cells into the islet

Our analysis of controllable beta-cell death in the excitotoxicity model suggested that beta-cell death leads to the expression of genes involved in inflammation and the crosstalk with the immune system. We wanted to test whether beta-cell death under physiological conditions also modulates inflammation and immune-cell crosstalk. Therefore, we turned to analyzing the role of developmental beta-cell death in shaping the immune cell component of islets. To this end, we quantified the number of islet-infiltrating macrophages and their inflammatory status in 6-month-old *Tg(ins:p35)* animals and WT siblings. We chose this late developmental stage as we wanted to assess the impact of blocking beta-cell death on the islet's immune system in the long term. Firstly, we characterized the islet resident macrophages and their inflammatory phenotype. Therefore, we took advantage of a double transgenic reporter line *Tg(mpeg1:mCherry),TgBAC(tnfα:EGFP)* and crossed it to our *Tg(ins:p35)* line to generate triple transgenic line *Tg(ins:p35); Tg(mpeg1:mCherry);TgBAC(tnfα:EGFP)*. The *Tg(mpeg:mCherry),TgBAC(tnfα:EGFP)* double transgenic line was used as a control. *Tg(mpeg1:mCherry)* transgenic line is widely used to label macrophages in zebrafish (Ellett et al, 2011) while *TgBAC(tnfα:EGFP)* is used to label cells that express the pro-inflammatory cytokine tumor necrosis factor-alpha (tnfα) (Marjoram et al, 2015). We observed that there was a striking reduction in the number of macrophages of *Tg(ins:p35)* animals as compared to clutch mate WT controls (Fig. 6A,B). Interestingly, we found that a significant percentage of macrophages in the islet of WT animals were expressing TNF-α (Fig. 6A'). However, the islets of *Tg(ins:p35)* did not harbor any pro-inflammatory macrophages (Fig. 6C). Therefore, we conclude that the genetic inhibition of beta-cell death leads to a significant reduction in islet resident macrophages, including the subset of pro-inflammatory macrophages. Since the *mpeg1* promoter has been shown to also mark B-cells in zebrafish (Ferrero et al, 2020), we generated a line expressing BFP under the macrophage *mfap4* promoter (Walton et al, 2015) and found that the majority of *mpeg1*-positive cells in the islets are also *mfap4*-positive, with reduced numbers in the islets of p35 animals, thereby confirming our conclusions (Appendix Fig. S10).

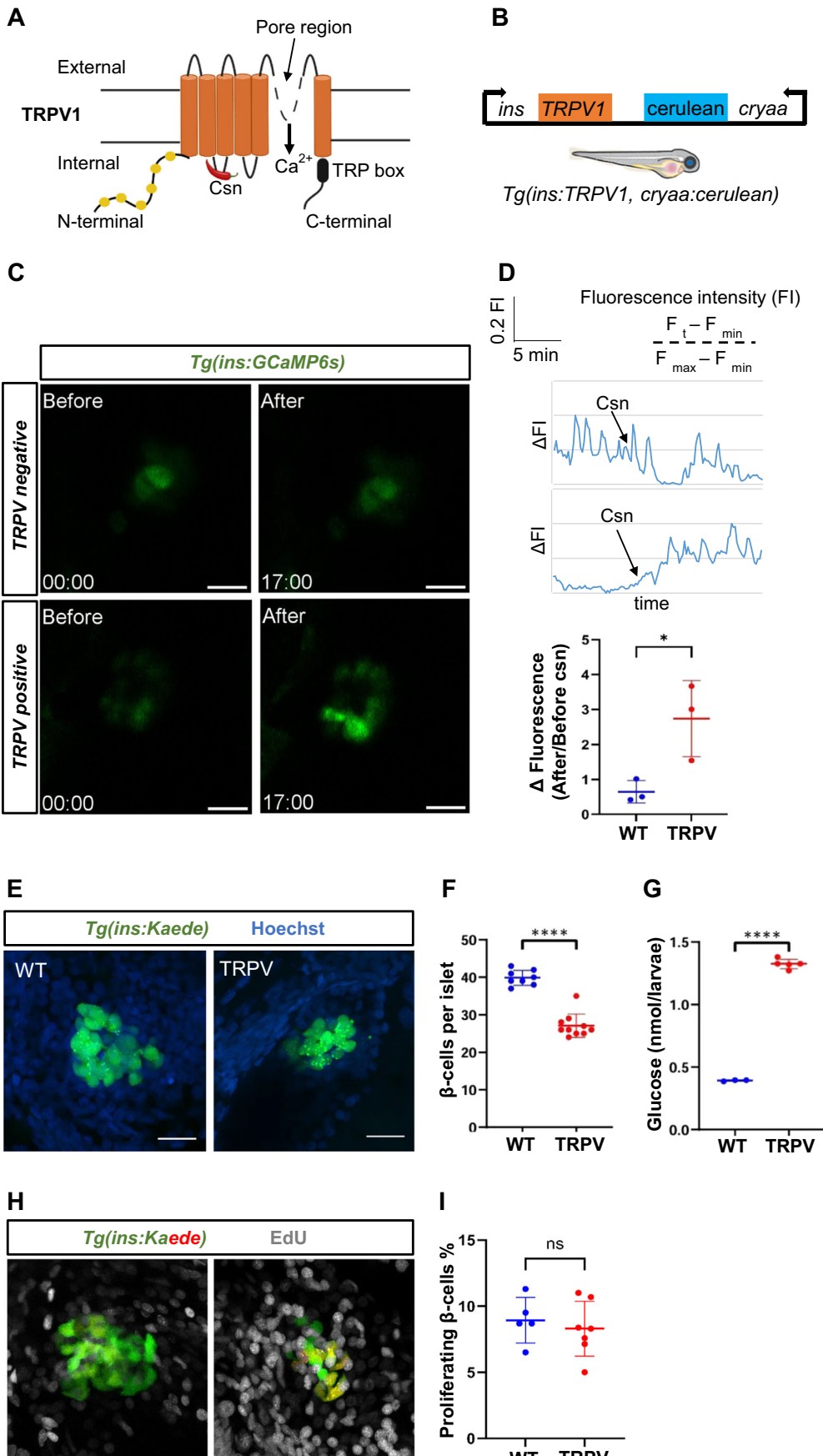

**Figure 4.  Generation of a genetic model of beta-cell excitotoxicity in zebrafish.**

(A) Structural representation of the transient receptor potential channel vanilloid channel (TRPV) and its mechanism of activation by the small molecule, capsaicin.
(B) Schematic representation of the genetic construct used to generate the transgenic model of beta-cell $Ca^{2+}$ excitotoxicity in zebrafish. The rat *TRPV* was cloned under the zebrafish insulin promoter. Cerulean expression under the control of the crystalline (*cryaa*) promoter serves as a marker of transgenic animals (blue eyes).
(C) Representative snapshots from live imaging of larvae expressing GCaMP6s in the beta cells. The images depict beta-cell GCaMP6s fluorescence before and after capsaicin (csn) stimulation of control and TRPV-expressing beta cells. Scale bar, 20 μm. $n = 3$ independent samples per group. The time-stamp indicates minutes.
(D) Traces and quantification of GCaMP6s fluorescence intensity over time for the islets shown in C. While the control larvae showed no perturbation in calcium dynamics, there was a sustained increase in calcium levels of the beta cells from *Tg(ins:TRPV)* larvae post csn-stimulation. The dot-plot graph shows the relative change in GCaMP6s fluorescence intensity after csn-stimulation in 5 dpf WT and TRPV larvae. Error bars are mean ± SD from $n = 3$ independent samples per group. Two-tailed *t*-test, *$p = 0.0331$. (E) Confocal images (maximum projection) of islets from *Tg(ins:Kaede)* and *Tg(ins:TRPV);Tg(ins:Kaede)* larvae following incubation with csn from 3 to 5 dpf. *Tg(ins:kaede)* marks the beta cells (green) while the nuclei are stained with Hoechst (blue). Scale bar 20 μm. (F) Quantifications of the number of beta cells per islet in control and *Tg(ins:TRPV)* larvae. Error bars are mean ± S D from $n = $ at least 7 independent samples per group. The horizontal bar represents the mean value. Unpaired two-tailed *t*-test with Welch's correction, ****$p = 0.0000000219$. (G) Plot showing average glucose value in WT and *Tg(ins:TRPV)* larvae following csn treatment for 48 h. Each dot corresponds to a pool of 10 larvae. Error bars are mean ± SD from $n = $ at least 3 independent replicates per group. Unpaired two-tailed *t*-test with Welch's correction, ****$p = 0.000000351$. (H) EdU-stained representative confocal images (maximum projection) of islets from *Tg(ins:Kaede)* and *Tg(ins:TRPV);Tg(ins:Kaede)* after photoconversion and treatment with csn from 3 to 5 dpf. The newly formed beta cells are marked with green; the pre-existing beta cells are in red/yellow, and EdU-positive cells are marked in gray. Scale bar 20 μm. (I) Quantification showing the percentage of proliferating (red/yellow) beta cells during csn treatment in both control and *Tg(ins:TRPV)*. The horizontal bar represents the mean value. Error bars are mean ± SD from n = at least 5 independent samples per group. Unpaired two-tailed *t*-test with Welch's correction, ns not significant (0.359). Source data are available online for this figure.

The results above made us wonder how the changes in the immune cell population influence the inflammatory profile of the beta cells. In order to assess beta-cell inflammation, we took advantage of *Tg(NF-kB:EGFP)* reporter line. We crossed *Tg(ins:p35)* transgenic line to *Tg(NF-kB:EGFP);Tg(ins:mCherry)* reporter line to generate *Tg(ins:p35);Tg(NF-kB:EGFP);Tg(ins:mCherry)* triple transgenic line for this experiment. Beta cells isolated from 6-month-old WT and *Tg(ins:p35)* animals were FACS sorted and analyzed for relative GFP expression (Fig. 6D). We plotted the distribution of GFP expression levels in beta cells to reveal heterogeneity among the beta cells. We segregated the beta cells into two different groups, which were labeled as NF-kB:GFP$^{hi}$ and NF-kB:GFP$^{low}$. We found that there was an increase in the percentage of beta cells that showed lower expression of GFP in *Tg(ins:p35)* animals as compared to age-matched WT controls (Fig. 6E).

## Tregs act to suppress islet inflammation

Given the decrease in NF-kB signaling in the islet upon macrophage reduction, we asked if there are changes in additional immune cells in the islet that may control inflammation. We focused on regulatory T cells (Tregs), which are known to suppress islet inflammation in type 1 diabetes (Barzaghi et al, 2012). First, we assessed the presence of Tregs in both the normal islet and the islet with beta-cell-specific caspase inhibition. To this end, we examined a *TgBAC(foxp3a:tagRFP)* reporter line (Hui et al, 2017) in which TagRFP is expressed under the control of a bacterial artificial chromosome (BAC) containing *foxp3a* regulatory elements. Foxp3 is a member of the Forkhead-box/winged-helix transcription factor family, which is necessary for the development and immunosuppressive function of regulatory T cells (Fontenot et al, 2005). We crossed the *TgBAC(foxp3a:tagRFP)* line with the *Tg(ins:p35)* line to generate a *Tg(ins:p35); TgBAC(foxp3a:tagRFP)* double transgenic line. We found that Tregs resided in the islets of control fish. However, there was an increase in the numbers of RFP cells infiltrating the pancreatic islet of *Tg(ins:p35)* animals as compared to clutch mate WT animals (Fig. 7A,B), indicating that beta-cell death inhibition promotes an anti-inflammatory environment in part by stimulating an increase in the Treg population.

To directly examine whether Tregs are involved in controlling beta-cell NF-kB activation, we investigated the inflammatory profile of zebrafish beta cells in 2-month-old *foxp3a*-deficient zebrafish (Sugimoto et al, 2017). Since the foxp3 transcription factor is crucial for the differentiation and immunosuppressive function of Tregs, these fish lack functional Tregs (Hui et al, 2017). We analyzed *foxp3a* mutants in the *Tg(ins:mCherry);Tg(NF-kB:EGFP)* transgenic background. Beta cells were dissociated from the primary islet of 2-month-old WT and *foxp3a$^{-/-}$* animals and FACS sorted (Fig. 7C). Subsequently, the beta cells were analyzed for relative GFP expression. We plotted the distribution of GFP-expression levels in beta cells. We segregated the beta cells into two different groups which were labeled as NF-kB:GFP$^{hi}$ and NF-kB:GFP$^{low}$. We found a significant increase in the percentage of NF-kB:GFP$^{hi}$ beta cells in Treg deficient mutants, as compared to WT siblings (Fig. 7D), indicating a premature activation of NF-kB signaling. Therefore, our data shows that Tregs play an active role in suppressing islet inflammation. Altogether, we show that beta-cell death shapes the islet's immune cells, which in turn feeds back to regulate the inflammatory state of the islet.

# Discussion

Our work shows that the onset of beta-cell death during juvenile stage plays a dual role. Firstly, it is crucial for the establishment of proper beta-cell mass, in the absence of which the islet size expands dramatically. Secondly, it plays an important role in orchestrating the islet's immune system component.

The relative contribution of cell death towards normal tissue development has been underappreciated. Dying cells are known to display morphological characteristics (Kerr et al, 1972). However, they are believed to be rapidly removed from the tissue making it difficult to assess the true extent of developmental cell death. For example, it has been shown previously in *Drosophila* that during epidermal cell turnover, apoptotic cells are rapidly removed by circulating haemocytes. While the mechanical remodeling i.e., apical constriction and extrusion of the dying epidermal cells, takes around 2 h, the time taken by the circulating haemocytes to completely phagocytose the dying epidermal cell is less than

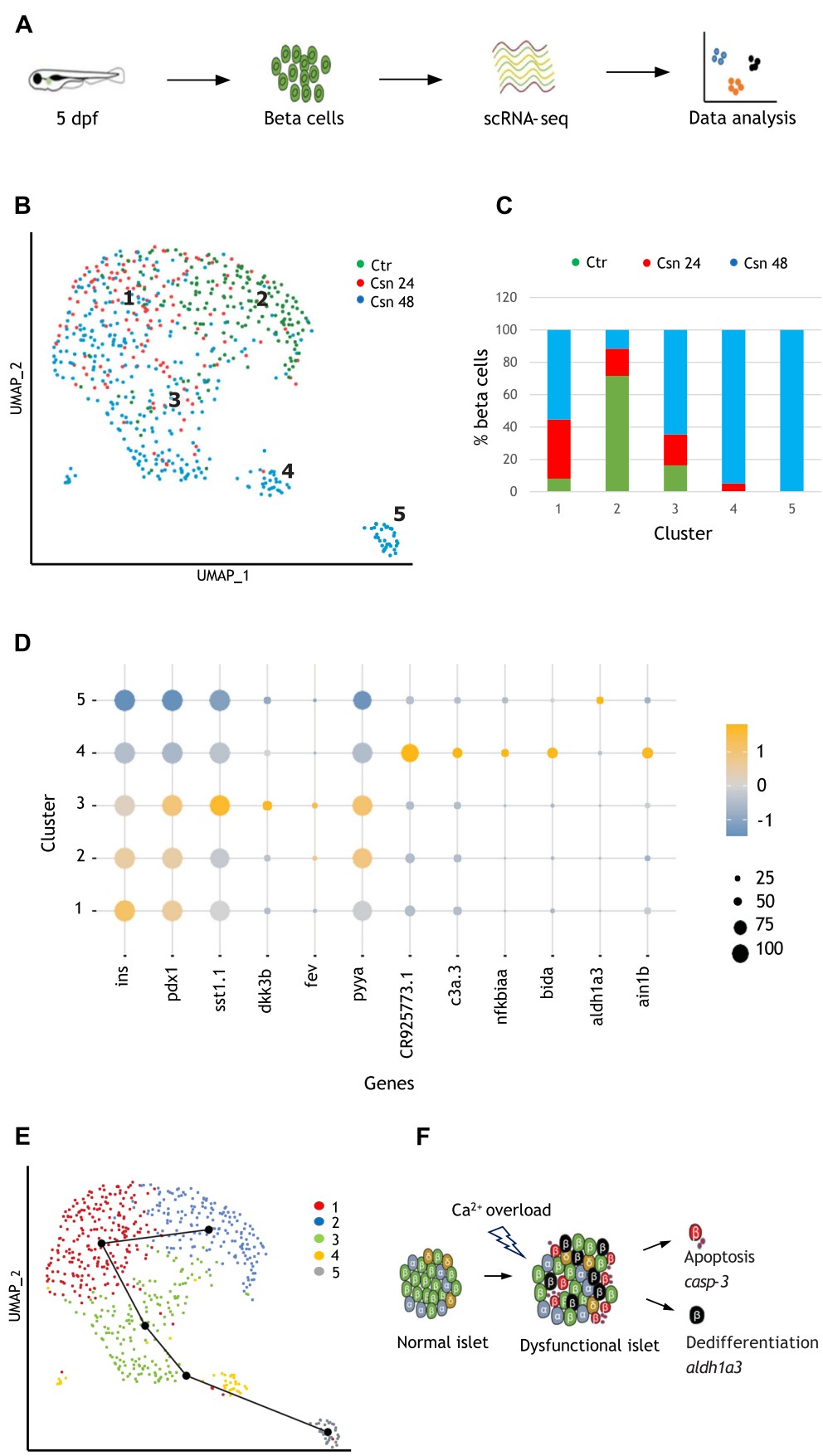

**Figure 5. Beta cells experiencing Ca²⁺ excitotoxicity show transcriptional trajectories toward cell death or dedifferentiation.**

(A) Schematic of the experimental design. (B) Uniform manifold approximation and projection plot depicting clusters of beta cells from all three samples. Clusters are numbered and color-coded according to the legend shown at the right. (C) The color-coded bar graph represents the cluster-wise distribution of beta cells from the three samples. (D) Dot plot indicating the expression levels and the percentage of cells expressing selected genes in each of the five beta-cell clusters. The size of the dot represents the percentage of cells expressing the gene in a particular cluster, while the color scale represents average expression of the gene in the cluster after scaling. (E) Beta-cell trajectory as suggested by pseudotime analysis with slingshot. The line connects the beta-cell clusters starting from 2, 1, 3, 4, and 5. (F) A schematic representation of our model based on single-cell transcriptomics. When experiencing chronic Ca²⁺ excitotoxicity, beta cells can undergo a reduction of beta-cell markers such as *ins* and *pdx1*. Some beta cells succumb to the stress, expressing cell death genes such as *casp-3* (cluster 4), while others go further down the dedifferentiation path, expressing *aldh1a3* and evading cell death (cluster 5).

10 min (Ninov et al, 2007). Furthermore, it has been estimated that in the kidney, more than 50% of the cells die during development with only 3% of them being identified by histological assessment (Coles et al, 1993). In this study, using a mathematical model and genetic inhibition of macrophages, we showed that the onset of beta-cell death during the juvenile stage is immediately associated with macrophage recruitment to the islets, consistent with previous findings showing that macrophages appear rapidly in 3–6 h in response to beta-cell damage in the ins:NTR model (Kulkarni et al, 2018). Similarly, macrophage accumulations were detected in islets from organ donors with T1D, suggesting that an onset of beta-cell death has attracted these macrophages. Donors positive for one or two T1D autoantibodies but not yet diagnosed showed a tendency towards more macrophages compared to controls, however this trend did not reach significance, warranting further investigation with a bigger cohort. A similar trend has already been observed for all CD45⁺ monocytes, CD4⁺ and CD8⁺ T-cells recruited into islets, and immune cell migration was much higher in double, compared to single AAb⁺ donors (Geravandi et al, 2021). Of note, these double AAb⁺ donors appear to show a higher number of macrophages than donors with only one single AAb in this study. Although here, we could access only a very low number of pancreases, it seems that CD68⁺ macrophages are the first that are attracted to the islets upon beta-cell death.

An important aspect of our study was the timing of developmental beta-cell death. Here, we showed that the onset of beta-cell death takes place by 20 dpf, a stage that coincides with the changes in the feeding behavior of zebrafish. During early larval stages i.e., until 5 to 7 dpf, zebrafish derive their nutrients from the yolk sac. Thereafter, they are fed on paramecium until approximately 15 dpf, followed by artemia (*Artemia nauplii*), which is rich in calories (Paffenhofer, 1967; Farias and Certal, 2016). These transitions have been linked to dynamic changes in beta-cell proliferation, with the feeding-fasting state having a dramatic impact on cell cycle progression (Ninov et al, 2013). We speculate that in addition to proliferation, the higher calorie diet leads to metabolic stress, which subsequently elicits the beta-cell turnover process during the juvenile stage of zebrafish. Similarly, in humans, it could be that a shift in dietary regimen from breastmilk to solid food in early childhood could cause metabolic stress on the beta cells leading to early beta-cell death. Therefore, a good question for the future will be to assess the impact of a high-calorie diet on the beta-cell turnover process, specifically during the early stages of pancreas development in humans.

The scope and relevance of our data extend beyond the zebrafish model as it corroborates previous observations suggesting that similar changes in beta-cell turnover also take place in mammals. It has been proposed that in rodents, a wave of developmental beta-cell death takes place during the early juvenile stage, i.e., from 14 to 17 days post-birth (Finegood et al, 1995; Scaglia et al, 1997). A recent study also documented extensive elimination of acinar cells in the mouse and human pancreas (Stolovich-Rain et al, 2023). Importantly in the non-obese diabetic (NOD) mice, the proposed wave of developmental beta-cell death coincided with the initiation of insulitis in the islet (Hoglund et al, 1999; Trudeau et al, 2000). While the exact mechanism that leads to the breakdown of immune tolerance to beta-cell-derived proteins is largely unknown, dysregulation in the apoptotic machinery and/or inefficient clearance of apoptotic cells due to genetic risk could trigger autoimmune processes (Vives-Pi et al, 2015). For instance, inefficient clearance of apoptotic beta cells could result in secondary necrosis where the beta-cell antigens could be taken up antigen-presenting cells and presented to T cells in the draining pancreatic lymph node. In humans, a longitudinal assessment of blood sugar levels and islet antibody development in infants with high genetic risk of T1D revealed that there was an increase in postprandial blood glucose levels before seroconversion (Warncke et al, 2022). The increase in postprandial glucose levels before seroconversion argues that beta-cell insult or defects during the early stages of development could drive this phenomenon. Early insult on beta cells of genetically predisposed individuals due to a plethora of factors such as environmental pollutants, viral infection, and metabolic stress may cause dysglycemia with subsequent initiation of autoimmunity (Donath, 2022). This is supported by the fact that immunotherapies, at best, delay the progression of autoimmunity without remission of T1D (Atkinson et al, 2011). In this study, using a mathematical model and genetic inhibition of macrophages, we showed that the onset of beta-cell death during the juvenile stage facilitates macrophage recruitment to the islet. Consistently genetic inhibition of caspase in our model led to beta-cell expansion in juvenile zebrafish with subsequent reduction of pro-inflammatory macrophages in the islet. Similarly, it has been shown that mice lacking caspase 3 were protected from developing diabetes in multiple low-dose streptozotocin (MLDS) diabetes models. This was accompanied by a complete absence of islet-infiltrating lymphocytes (Liadis et al, 2005). Here, we detected an increase in macrophage accumulations in islets from donors with type 1 diabetes, suggesting that an early onset of cell death has attracted these macrophages. Hence, the early intervention strategy to prevent excessive beta-cell apoptosis could be a means to inhibit excessive pro-inflammatory immune cells from infiltrating the pancreatic islet and prevent diabetes development in genetically susceptible individuals.

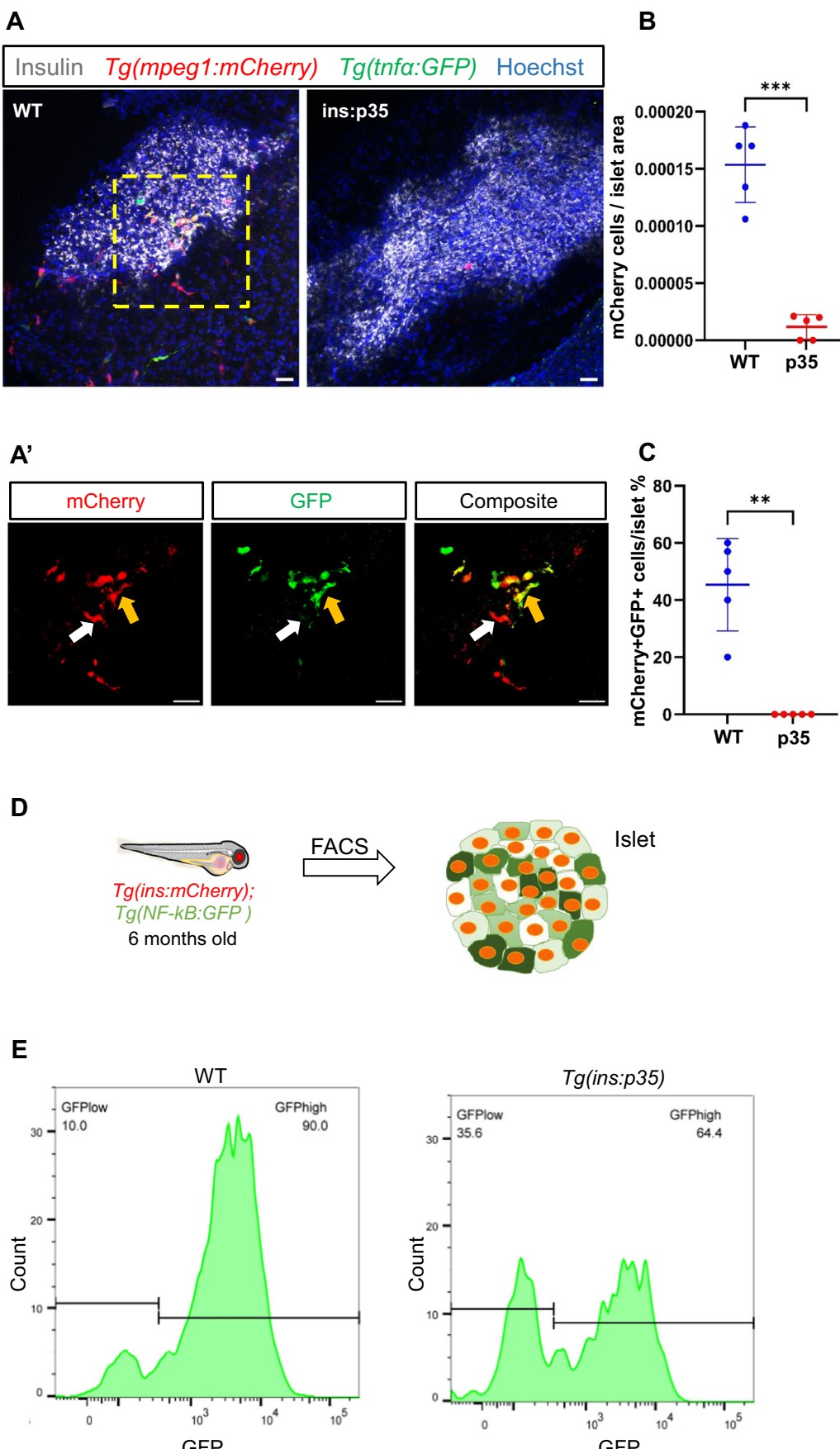

**Figure 6. Beta-cell death regulates the islet's immune component and inflammation.**

(A) Representative confocal images (maximum projection) of primary islets from 6 months old WT and *Tg(ins:p35)* animals in the transgenic background of *Tg(mpeg:mCherry);Tg(tnfa:GFP)*. Immunostaining against insulin is used to label the beta cells (gray). *Tg(mpeg:mCherry)* reporter marks the macrophages (red), *Tg(tnfa:GFP)* marks the tnf-α expressing cells (green) while the nuclei are stained with Hoechst (blue). Scale bar 20 μm. (A') Insets show high-magnification single planes from the confocal projection of the WT animal (corresponding to the area marked using a yellow dotted line in A). Scale bar, 20 μm. A subset of macrophages express tnf-α (yellow arrow) while others do not (white arrow). (B) Quantification showing the number of mCherry-positive cells per unit islet area in WT and *Tg(ins:p35)*. Error bars are mean ± SD from $n = 5$ independent samples per group. The horizontal bar represents the mean value. Unpaired two-tailed *t*-test with Welch's correction, ***$P = 0.000494$. (C) Quantification showing the percentage of mCherry and tnfα double-positive cells in the islet of WT and *Tg(ins:p35)* animals. Error bars are mean ± SD from $n = 5$ independent samples per group. The horizontal bar represents the mean value. Unpaired two-tailed *t*-test with Welch's correction, **$P = 0.003271$. (D) Beta cells from 6 months old *Tg(ins:mCherry);Tg(NF-kB:GFP)* and *Tg(ins:p35);Tg(ins:mCherry);Tg(NF-kB:GFP)* animals were analysed using FACS. The data shows the results from the analysis of islets from $n = 5$ combined fish per group with two biological replicates. (E) The FlowJo graph shows GFP intensity (along the X-axis) and the distribution of beta cells from WT and *Tg(ins:p35)* animals, respectively. Horizontal lines indicate the division point between GFP$^{low}$ and GFP$^{high}$ levels. Percentage values represent the proportion of cells with GFP$^{low}$ or GFP$^{high}$ expression. Source data are available online for this figure.

Another important aspect of islet immunity is the degree of beta-cell inflammation. We observed that there was a reduced proportion of beta cells with high NF-kB activity in the islet of p35-expressing animals. This could be due to cell-intrinsic or extrinsic factors. For instance, it has been previously demonstrated that overexpression of IL-1β in the beta cells leads to robust activation of NF-kB in the beta cells of zebrafish larvae (Delgadillo-Silva et al, 2019). Furthermore, overexpression of *tnfrsf1b* in the beta cells leads to premature activation of NF-kB in the beta cells of zebrafish larvae (Janjuha et al, 2018). p35 is a pan-caspase blocker, i.e., it can block caspases that regulate cell death as well as caspases that regulate inflammation, such as caspase 1. However, our subsequent analysis of *Tg(il1b:EGFP)* and *Tg(tnfα:GFP)* reporter lines failed to identify expression of IL-1β or TNFα in the beta cells of WT animals, implying that at the basal level, beta cells do not express these pro-inflammatory cytokines (Appendix Fig. S11). Alternatively, tissue-resident immune cells such as macrophages are known to regulate tissue inflammation by secreting different cytokines such as TNFα and IL-1β. For example, it has been shown in zebrafish that with advancing age, there is an accumulation of pro-inflammatory macrophages that express TNFα, which regulates NF-kB activity in the beta cells (Janjuha et al, 2018). Furthermore, in rodents with increasing age, there is an accumulation of IL-1β expressing CD45+ islet immune cells, which negatively impacts beta-cell function and proliferation with myeloid cell-specific deletion of IL-1β preserving beta-cell function and proliferation in aged (1-year-old) mice (Boni-Schnetzler et al, 2021).

Besides macrophages, Tregs are known to play a crucial role in regulating tissue homeostasis and immunological self-tolerance (Shevach, 2009). FOXP3, a member of the forkhead/winged-helix family of transcription factors, plays a pivotal role in the development and suppressive function of Tregs (Fontenot et al, 2005). Loss of function mutation in the *Foxp3* gene (scurfy mice) leads to a fatal autoimmune disease characterized by hyperactive CD4 + T cells, and the animals succumb to death within 4 weeks of age (Lyon et al, 1990; Singh et al, 2007). In humans, mutations in the *FOXP3* gene lead to X-linked autoimmune lymphoproliferative disorder called IPEX (Immune dysregulation Polyendocrinopathy Enteropathy X-linked) syndrome, which includes insulin-dependent diabetes (Bennett and Ochs 2001). Furthermore, *foxp3a* mutant zebrafish show excessive infiltration and proliferation of lymphocytes in different organs such as the gill, intestine, and pharynx (Sugimoto et al, 2017). Here, we showed using an inflammation reporter that *foxp3* deficiency leads to exacerbated islet inflammation.

An important question for the field of diabetes research is whether beta cells are equally susceptible to stress or whether there is an underlying heterogeneity i.e., some beta cells are more vulnerable than others. Indeed, it has been shown that even in patients with long-standing T1D, some beta cells persist in the islet, albeit their number is significantly reduced (Meier et al, 2005). This could be due to continuous beta-cell regeneration or to the ability of some beta cells to evade the immune assault. It is known that with advancing age, beta-cell proliferation decreases dramatically (Perl et al, 2010), implying that the residual beta cells are able to escape immune-mediated destruction. Based on the remarkable heterogeneity among the beta cells, it could be that a subset of resilient beta cells is able to withstand chronic stress. However, the underlying heterogeneity among the beta-cells' response to stress is not well understood. One possible therapeutic approach would be to harness this heterogeneity and decode which factors help withstand the stress. A recent study showed that zMIR (zebrafish skeletal muscle insulin resistance) zebrafish fed on a high-fat diet develop chronic beta-cell ER stress leading to a marked reduction in the number of beta cells (Yang et al, 2020). This wave of beta-cell destruction is orchestrated by a coordinated response of macrophages and neutrophils towards beta-cell stress. Interestingly, only those beta cells are destroyed by neutrophils that were visited previously by macrophages, raising the possibility that some beta cells are more vulnerable to immune-mediated destruction than others (Yang et al, 2022). However, the underlying heterogeneity among the beta cells was not investigated, as to why some beta cells are destroyed by the immune cells while others escape the destruction. In the chemogenetic model of beta-cell stress, we show that there is remarkable transcriptional heterogeneity and found that some beta cells are more vulnerable than others. A subset of beta cells expressed genes related to cell death and inflammation such as *casp-3* and *nfkbiaa*. In addition, this group of beta cells expressed genes related to immune system activation such as the complement complex *c3a.3* and *c3b.1*. Genetic inhibition of *c3a* has been shown to dramatically reduce the migration of neutrophils to the site of injury in zebrafish (Houseright et al, 2020). This shows that during chronic stress, a subset of vulnerable beta cells activates the immune system and progresses toward inflammatory cell death. In contrast, we found that a subset of beta cells expresses the zebrafish orthologue of *aldh1a3*, which is a canonical marker for dedifferentiating beta cells in mammals. It has been shown previously using a mouse model of diabetes that there is a striking upregulation of ALDH1A3 in a subset of dedifferentiating beta cells (Kim-Muller et al, 2016). Furthermore, there is a

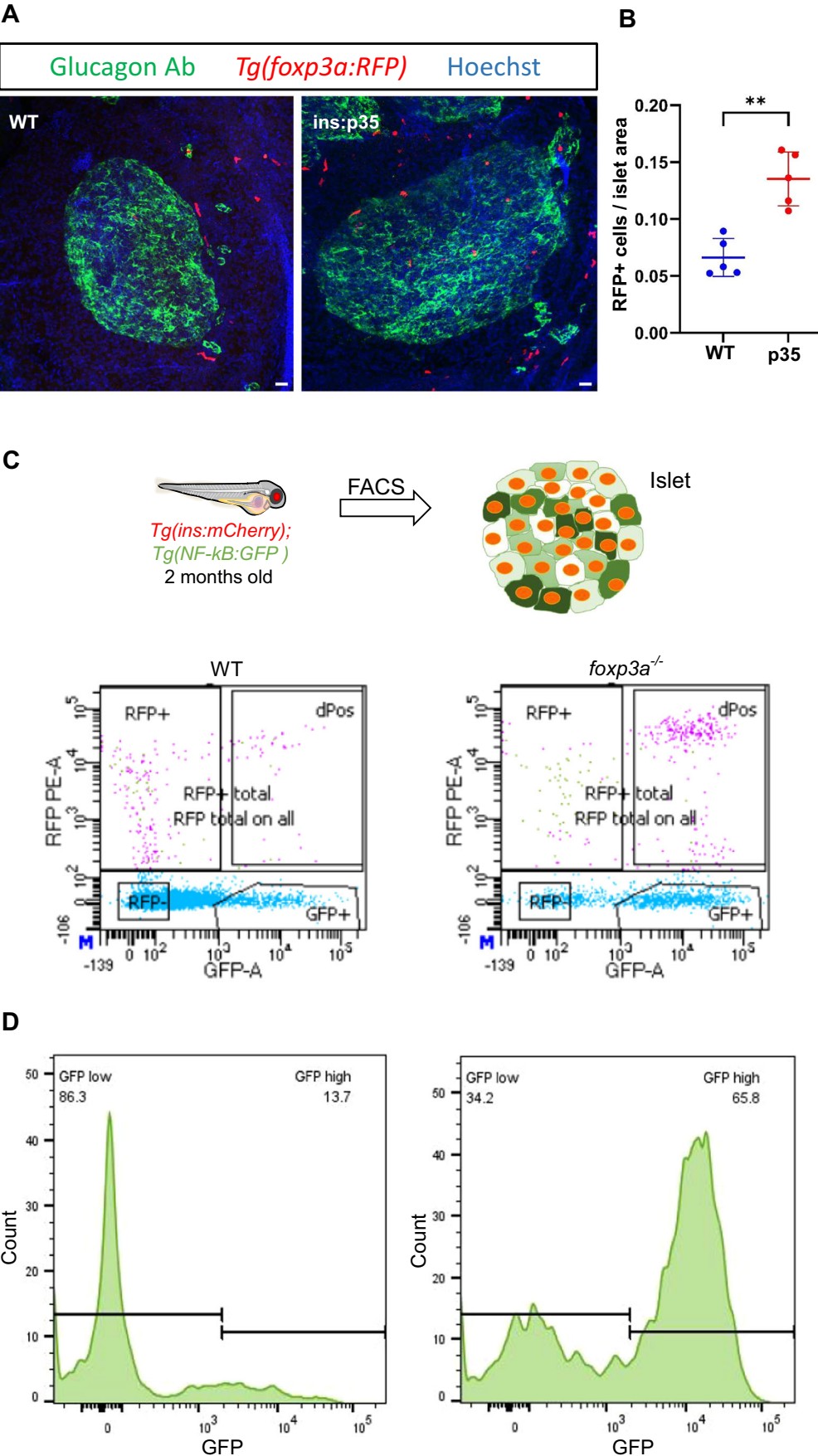

**Figure 7. Zebrafish Tregs populate the islets of p35-expressing beta cells and regulate the levels of islet inflammation.**

(A) Representative confocal images (maximum projection) of primary islets from 6 months old WT and *Tg(ins:p35)* animals in the transgenic background of *Tg(foxp3a:RFP)*. Immunostaining against glucagon marks the alpha cells (in green), *Tg(foxp3a:RFP)* marks the Tregs (in red), while Hoechst staining was done to label the nuclei (blue). Scale bar, 20 μm. (B) Quantification showing the number of RFP positive cells per unit islet area in WT and *Tg(ins:p35)*. Error bars are mean ± SD from $n = 5$ independent samples per group. The horizontal bar represents the mean value. Unpaired two-tailed *t*-test with Welch's correction, $**P = 0.001$. (C) Schematic showing the experimental setup for FACS. Beta cells from 2 months old WT and *foxp3$^{-/-}$* mutants in the *Tg(ins:mCherry);Tg(NF-kB:GFP)* transgenic background were analysed using FACS for GFP expression. The data shows the results from the analysis of islets from $n = 5$ combined fish per group from two biological replicates. (D) FlowJo graph shows GFP intensity (along the X-axis) and the distribution of beta cells from WT and *foxp3a$^{-/-}$* animals, respectively. Horizontal lines indicate the division point between GFP$^{low}$ and GFP$^{high}$ levels. Percentage values represent the proportion of cells with GFP$^{low}$ or GFP$^{high}$ expression. Source data are available online for this figure.

significant increase (>3-fold) in ALDH1A3 expression in the islets of subjects with type 2 diabetes as compared to healthy individuals (Cinti et al, 2016). In line with these findings, our transcriptomic dataset showed that a subset of stressed beta cells upregulated *aldh1a3*, indicating that excitotoxicity might cause beta-cell dedifferentiation. We cannot exclude the possibility that those beta cells that go further down the dedifferentiation path also reduce TRPV expression helping them to alleviate the stress. In the future, it will be important to assess whether beta-cell dedifferentiation also happens during the natural development of the pancreas and contributes to beta-cell loss. In an elegant study published recently, pharmacological and genetic inhibition of ALDH1A3 in diabetic mice helped lower blood glucose levels and increase insulin secretion, indicating that targeting ALDH1A3 might offer a therapeutic means to treat beta-cell dysfunction (Son et al, 2023). Taken together, our study provides significant insight into developmental beta-cell death, its role in orchestrating the crosstalk with the islet's immune cells, and the underlying heterogeneity in beta-cell's response to chronic stress which could be relevant for further understanding of T1D.

# Methods

### Reagents and tools table

| Reagent/resource | Reference or source | Identifier or catalog number |
| --- | --- | --- |
| **Experimental models** | | |
| **Zebrafish lines** | | |
| WT | AB | ZDB-GENO-960809-7 |
| Tg(ins:p35, cryaa:mCherry) | This paper | |
| Tg(ins:TRPV1, cryaa:cerulean) | This paper | |
| Tg(insulin:loxP:mCherrySTOP:loxP:H2B-GFP)s934 | Ninov et al, 2013 | ZDB-ALT-111031-2 |
| Tg(ins:kaede)jh6 | Pisharath et al, 2007 | ZDB-ALT-070606-4 |
| Tg(mpeg1:EYFP-NTR)w202 | Petrie et al, 2014 | ZDB-TGCONSTRCT-140903-2 |
| Tg(ins:GCaMP6s;cryaa:mCherry)tud202 | Singh et al, 2017 | ZDB-ALT-181221-2 |
| irf8st96 | Shiau et al, 2015 | ZDB-TALEN-150428-1 |
| Tg(mpeg1:mCherry)gl23 | Ellett et al, 2011 | ZDB-ALT-120117-2 |
| Tg(NF-kB:EGFP)nc1 | Kanther et al, 2011 | ZDB-TGCONSTRCT-120409-6 |

| Reagent/resource | Reference or source | Identifier or catalog number |
| --- | --- | --- |
| TgBAC(tnfa:GFP)pd1028 | Marjoram et al, 2015 | ZDB-ALT-150603-5 |
| Tg(Il1b:GFP)sh445 | Ogryzko et al, 2019 | ZDB-ALT-190307-8 |
| TgBAC(foxp3a:TagRFP,cryaa:EGFP)vcc3 | Hui et al, 2017 | ZDB-ALT-180515-10 |
| foxp3avcc6 | Sugimoto et al, 2017 | ZDB-ALT-170906-5 |
| **Recombinant DNA** | | |
| **Antibodies** | | |
| anti-EGFP | Abcam | ab13970 |
| anti-CD68 | Dako | M0814 |
| anti-glucagon | Sigma | G2654 |
| anti-insulin | GeneTex | 128490 |
| anti-insulin | Dako | A0546 |
| anti-Pdx1 | Chris Wright Lab | House-made |
| anti-tRFP | Rabbit, Evrogen | AB233 |
| Alexa Fluor 568 anti-rabbit | Invitrogen | A-21428 |
| Alexa Flour 488 anti-mouse | Invitrogen | A-28175 |
| Alexa Fluor 647 anti-rabbit | Invitrogen | A-21244 |
| Alexa Fluor 488 anti-chicken | Invitrogen | A-11039 |
| **Oligonucleotides and other sequence-based reagents** | | |
| pDEST mfap4:turquoise2 vector | Addgene | 135218 |
| Zebrafish genome | GRCz11 | www.ensemble.org |
| **Chemicals, enzymes, and other reagents** | | |
| Capsaicin | Sigma | M2772 |
| Dimethyl Sulfoxide | Sigma | D8418 |
| Metronidazole | Sigma | PHR1052 |
| Click-iT™ Plus EdU kit | Invitrogen | C10340 |
| VECTASHIELD® Mounting Medium | Vector Lab | 101098-042 |
| Paraformaldehyde | EM Grade | GF700000 |
| Sucrose | Sigma | S5391 |
| D-Glucose | Sigma | D7021 |
| Low-melting agarose | VWR | 89125-534 |
| Click-iT™ Plus TUNEL Assay Kit | Invitrogen | C10617 |
| HBSS (Hank's Balanced Salt Solution) | Invitrogen | 14175095 |
| Trypsin | ThermoFisher | 15090046 |
| Dispase II | Merck | SCM133 |
| RNAse Inhibitor | NEB | M0314 |
| Sera-Mag SpeedBeads | GE Healthcare | FIS11859912 |

| Reagent/resource | Reference or source | Identifier or catalog number |
|---|---|---|
| KAPA2G Robust HotStart Ready Mix | MERCK | KK5701 |
| Calcein Violet 450 AM Viability Dye | ThermoFisher | 65-0854-39 |
| Hoechst dye | ThermoFisher | 33342 |
| BioVision Glucose Assay Kit | Abcam | ab65333 |
| Blood glucose measurement ACCU check | Roche | 10119750 |
| GeneJET Plasmid Miniprep Kit | Thermofisher | K0502 |
| Primers and Sequencing | Microsynth | https://www.microsynth.com/home-ch.html |
| Plasmid Maxi Kit | QIAGEN | 12963 |
| **Software** | | |
| Fiji/ImageJ | | https://imagej.net. |
| Imaris Bitplane | Imaris | |
| GraphPad Software 7.0 | GraphPad | |
| ZEN software (black) | Carl Zeiss AG | |
| Biorender | Biorender | |
| COPASI | Hoops et al, 2006 | |
| FitMultiCell | Alamoudi et al, 2023 | |
| Morpheus | Starruß et al, 2014 | |
| R | Seurat package 3.9 | |
| **Other** | | |
| Photoconversion LED | Loctite | 1359255 |

## Ethical statement

All animal experiments were carried out following the animal welfare guidance and under the supervision of the Landesdirektion Sachsen, Germany (permits: TVV 45/2018, TVV8/2022, TVV17/2023, TVV64/2023, Tv vG 7/2024, TVV 48/2024, T12/2016, T15/2023, and all corresponding amendments). All efforts were made to strictly abide by the 3R principle (to avoid animal suffering and to limit the number of animals used per experiment). Whenever possible, experiments were performed with zebrafish larvae that did not exceed the 5 days post fertilization (dpf) stage, as stated in the animal protection law (TierSchVersV §14). According to the EU directive 2010/63/EU, the use of these earlier zebrafish stages reduces the number of experimental animals, in agreement with the principles of the 3Rs.

## Zebrafish feeding regimen and selection for time course analysis of beta-cell mass

Adult *Tg(ins:p35)* animals were set for mating with the WT AB strain in pair mating tanks. The next morning, the divider was removed to allow mating, and the resulting fertilized eggs were collected within 2 h in order to ensure synchronous development. The embryos were maintained in 90 mm petri dishes (60 embryos per plate) containing E3 medium at 28.5 °C. After 5 days post

fertilization, the larvae were set out to grow in the nursery, where they were fed externally in a regulated manner. From 5 till 15 dpf, they were fed with paramecia twice daily, followed by which they were fed with Artemia thrice daily. Two months old juvenile animals were fed with flake food in addition to Artemia. After the animals were 4 months old their diet became restricted to flake food, once daily.

For the time course assessment of beta-cell number, p35 and WT clutch mate animals were used. All possible measures were taken to compare the control and p35 fish at approximately the same standard body length in order to eliminate potential discrepancies in islet size that may arise due to relative differences in animal growth, which becomes variable at the juvenile stage (Parichy et al, 2009). The table depicts the average length of the animals used for this analysis at the appropriate stages of development.

| Developmental stage (dpf) | WT (mm) | Tg(ins:p35) (mm) |
|---|---|---|
| 5 | 3.9 ± 0.05 | 3.9 ± 0.08 |
| 15 | 5 ± 0.15 | 4.9 ± 0.12 |
| 20 | 7 ± 0.10 | 7.1 ± 0.09 |
| 23 | 7.7 ± 0.12 | 7.5 ± 0.08 |
| 27 | 8.3 ± 0.11 | 8.4 ± 0.09 |
| 30 | 9.1 ± 0.15 | 9.3 ± 0.15 |

## Zebrafish husbandry and strains

WT or transgenic zebrafish of the outbred AB strain were used in all experiments. Zebrafish were kept and bred under standard conditions at 28 °C. Animals were chosen at random for all experiments. Published transgenic lines used in this study were *Tg(insulin:loxP:mCherrySTOP:loxP:H2B-GFP)[s934]* (Singh et al, 2017), abbreviated *Tg(ins:mCherry); irf8[st96]* (Shiau et al, 2015), *Tg(ins:GCaMP6s;cryaa:mCherry)[tud202]* (Singh et al, 2017), *Tg(ins:kaede)[jh6]* (Pisharath et al, 2007), *Tg(mpeg1:EYFP-NTR)[w202]* (Petrie et al, 2014); *TgBAC(tnfa:GFP)[pd1028]* (Marjoram et al, 2015); *Tg(Il1b:GFP)[sh445]* (Ogryzko et al, 2019), *Tg(mpeg1:mCherry)[gl23]* (Ellett et al, 2011); *Tg(NF-kB:EGFP)[nc1]* (Kanther et al, 2011); *TgBAC(foxp3a:TagRFP,cryaa:EGFP)[vcc3]* (Hui et al, 2017), abbreviated as *Tg(foxp3a:RFP)* and *foxp3a[vcc6]* (Sugimoto et al, 2017), abbreviated as *foxp3a[−/−]*.

## Construction of Tg(ins:p35; cryaa:mCherry)

To generate the *ins:p35; cryaa:RFP* construct, the plasmid pEF-BOS-XC-p35 was PCR amplified, adding a Kozak sequence. During PCR, flanking 5′ *Eco*RI and 3′ *Pac*I sites were also introduced. A previously established plasmid backbone containing *ins:mAG-zGeminin; cryaa:mCherry* (Ninov et al, 2013) was digested with *Eco*RI/*Pac*I and the coding region of the p35 gene was ligated using the *Eco*RI/*Pac*I sites. The entire construct was flanked with I-*Sce*I sites to facilitate transgenesis. Several founders were screened and one founder with Mendelian segregation was selected. This line was used in all further experiments.

## Construction of Tg(ins:TRPV1-tagRFP; cryaa:mCerulean)

For the construction of the *Tg(ins:TRPV1-tagRFP; cryaa:mCer-ulean)* abbreviated as *Tg(ins:TRPV)* transgenic line, the TRPV1-RFPT was PCR amplified from *islet1:GAL4VP16,4xUAS:TRPV1-RFPT^ct825* (Chen et al, 2016). The PCR product was flanked by *ECoR*I and *Pac*I site. A previously established plasmid backbone with *ins: mKO2-zCdt1; cryaa: CFP* (Ninov et al, 2013) was digested with EcoRI/PacI and TRPV1-tagRFP was ligated using the EcoRI/PacI sites. The entire construct is flanked with I-SceI sites to facilitate transgenesis. I-SceI based transgenesis was performed, and founders were selected based on CFP (blue) fluorescence in the eye and red expression in the pancreatic beta cells.

## Generation of *Tg(mfap4: turquoise2)* line

To generate the *Tg(mfap4: turquoise2)* line, the *pDEST mfap4:tur-quoise2* vector purchased from Addgene (#135218) was injected in single-cell zebrafish embryos. The line was generated by using Tol2-mediated transgenesis (Suster et al, 2009). Several founders were selected based on Turquoise2 (blue) expression in macro-phages. The founder of Mendelian segregation was selected.

## Tissue collection and clearing

Juvenile and adult animals were killed by immersion in MS-222. The gut was dissected out and fixed in 4% paraformaldehyde for 48 h at 4 °C. Post fixation, the pancreas was dissected out from the gut and washed multiple times in PBS.

To facilitate confocal imaging of the islet, tissue clearing of the adult pancreas was performed according to the CUBIC-based clearance technique (Susaki et al, 2014).

## Cell counting

Total number of beta cells in the islets were counted using Imaris (Biplane). For counting beta cells in *Tg(ins:mCherry)* and *Tg(ins:p35); Tg(ins:mCherry)*, the "spots" function of Imaris with appropriate thresholding, was used to count all the red cells in stacks spanning the entire islet.

## Mathematical modeling

The model considers two subpopulations, one of proliferative beta cells with a time-dependent number $N_p(t)$ and another of quiescent beta cells with a time-dependent number $N_q(t)$, respectively. Proliferative cells, in the model, may reversibly switch to the quiescent state with a rate $k_i$ and back with a rate $k_a$, see Fig. 2A for all considered processes and their rate parameters. The cell cycle duration of proliferative beta cells is denoted $T_c$. We denote the rate of beta-cell neogenesis by $v$ and that of cell death by $\delta$. The two differential equations for numbers $N_p(t)$ and $N_q(t)$ are as follows.

$$\frac{dN_p(t)}{dt} = \frac{\ln 2}{T_C} N_p(t) - k_i N_p(t) + k_a N_q(t) + v\frac{N_p(t)}{N(t)} - \delta N_p(t)$$

$$\frac{dN_q(t)}{dt} = k_i N_p(t) + k_a N_q(t) + v\frac{N_q(t)}{N(t)} - \delta N_q(t)$$

The neogenesis process, with rate $v$, of the overall beta-cell mass (modeled by the second-to-last term in each equation) is scaled such that it does not disturb the ratio between both subpopulations and that the scaling factors vanish when both equations are added. This model predicts $Np(t)$ and $Nq(t)$ and then the sum $N(t) = Np(t) + Nq(t)$ is calculated and compared to the experimental cell number counts. As the considered time period of 25 days from 5 to 30 dpf spans part of the larval stage, which is rather homogeneous compared to the highly regulated embryonic period before, we consider all processes with constant rates except for cell death. Cell death is assumed to be absent initially, to start at time $T_{onset}$ and to vanish again after time $T_{onset} + T_{duration}$. Model fitting was performed with the two open-source software packages COPASI (Hoops et al, 2006) and FitMultiCell (Alamoudi et al, 2023). Results were verified by independent simulation in the open-source software package Morpheus (Starruß et al, 2014). The best fit of the 6 model parameters to the 66 individual experimental data points (as shown in Fig. 1E; Appendix Fig. S1B, and Fig. 2A') is obtained with the following parameter set.

| Parameter | Unit | Value fitted for WT | Value fitted for p35 |
|---|---|---|---|
| Cell cycle duration $T_C$ | d | 1.05 ± 0.09 | set to same as WT |
| Neogenesis flux $v$ | number of cells/d | 3.3 ± 0.5 | set to same as WT |
| Death rate $\delta$ | % of cells /d | 3.0 ± 0.6 | set to 0 |
| Death onset $T_{onset}$ | dpf | 16.3 ± 1.6 | set to same as WT |
| Death duration $T_{duration}$ | d | >14 | set to same as WT |
| Inactivation rate $k_i$ | 1/d | set to 1 | set to same as WT |
| Activation rate $k_a$ | 1/d | 0.03 ± 0.05 | set to same as WT |
| Initial proliferating $N_p(t = 0)$ | number of cells | set to 3 | set to same as WT |
| Initial quiescent $N_q(t = 0)$ | number of cells | set to 41 | set to same as WT |

Importantly, a cell death rate of 3.0 ± 0.6% per day from 16.3 ± 1.6dpf onwards is predicted in the WT condition and the dominance of $k_i$ over $k_a$ reflects the low fraction of PCNA[+] cells (Fig. 1F).

## Immunofluorescence and image acquisition

The samples were permeabilized in 1% PBT (Triton X-100) and blocked in 4% PBTB (BSA). Primary and secondary antibody staining were performed overnight at 4 °C. Primary antibodies used in this study were anti-insulin (rabbit, GeneTex 128490 and guinea pig, Dako#A0546) at 1:200, anti-glucagon (mouse, Sigma G2654) at 1:500, anti-tRFP (rabbit, Evrogen AB233), anti-L plastin (rabbit, custom made) at 1:1000 and anti-EGFP (chicken, Abcam ab13970) at 1:1000, anti-Pdx1 (gift from Dr. C. Wright, Vanderbilt University), mouse anti-CD68 (DAKO M0814) at 1:200. Secondary antibodies at 1:500 dilutions used in this study were Alexa Fluor

568 anti-rabbit, Alexa Fluor 568 anti-rabbit, Alexa Flour 488 anti-mouse, Alexa Fluor 647 anti-rabbit and Alexa Fluor 488 anti-chicken. Samples were mounted in Vectashield and imaged using a Zeiss LSM 780 or Nikon Ti MEA53200. The ZEN Black software or ImageJ were used to add scale bars and adjust brightness and contrast. PowerPoint was used for labeling. Human tissue samples were processed as previously described (Geravandi et al, 2021).

## In vivo calcium imaging

From 1 dpf onwards, *Tg(ins:GCaMP6s)* and *Tg(ins:TRPV)*; *Tg (ins: GCaMP6s)* transgenic zebrafish larvae were treated with 0.003% 1-phenyl thio-urea (PTU) (200 µM) to inhibit pigmentation. At 4.5 dpf, the larvae were anaesthetized using $0.4\,\text{g}\,\text{l}^{-1}$ Tricaine and mounted on their side in 1% low melting point agarose in a $35 \times 10$ mm culture dish. Live imaging was performed on an inverted laser scanning confocal system, ZEISS LSM 780, inverted with a 20X water immersion objective. The GCaMP6 and mCherry signals were acquired simultaneously using the 488 and 561 nm laser lines. Videos were recorded at a 10 s per image (0.1 Hz) frame rate unless indicated otherwise, with a $Z$-step thickness of 1.2 µm, covering on average 35 µm, and an $XY$ resolution of 0.12 µm per pixel ($512 \times 512$ pixels). We acquired images at 1 frame per second for 10 s before and 440 s after Csn administration. We added 1 mL of Csn at 10 µM to the side of the culture dish for a final concentration of 10 µM. We measured fluorescence intensity by drawing regions of interest around pancreatic islet region using ImageJ. The change in beta-cell fluorescence intensity ($F_I$) for the entire imaging period was normalized using the following equation: $(F_t - F_{min})/(F_{max} - F_{min})$, where $F_t$ is the fluorescence at a given time and $F_{max}$ and $F_{min}$ are the maximum and minimum fluorescence for the entire imaging period, respectively. The change in fluorescence was calculated by taking the ratio of average fluorescence before and after csn stimulation.

## EdU (5-Ethynyl-2′-deoxyuridine) labeling

To label proliferating cells in larval zebrafish, the larvae were placed in E3 media with 2.5 mM EdU for 48 h starting at 3 dpf. The larvae were fixed in 4% paraformaldehyde 24 h at 4 °C. After fixation, the larvae were rinsed with PBS and their skin was removed manually using forceps under the stereoscope. Next, the larvae were permeabilized using 1% Triton-X in PBS. Thereafter, EdU detection was performed according to the kit protocol Click-iT EdU Alexa Fluor 647 Imaging Kit (C10340 Fisher Scientific).

Juvenile WT and *Tg(ins:p35)* animals were incubated in 2 mM EdU solution for 24 h. After EdU treatment, the animals were killed and the gut was fixed in 4% paraformaldehyde for 24 h at 4 °C. Post fixation, the PFA solution was removed, and the gut was dissected manually to isolate the pancreas. The pancreatic samples were permeabilized using 1% Triton X in PBS. Subsequently, EdU detection was performed according to the kit protocol Click-iT EdU Alexa Fluor 647 Imaging Kit (C10340 Fisher Scientific).

## Photoconversion of Kaede

For photoconversion of Kaede positive beta cells, 3 dpf *Tg(ins:-Kaede)* larvae were photoconverted using Loctite LED flood system equipped with a 405-nm laser. The LED array evenly illuminated the entire petri dish containing larvae with an intensity of 400 mW/ cm² and was triggered using a foot pedal switch. An internal timer using an LED controller (Loctite, 1359255) was used for precise timing of light exposure. Each sample was given five pulses of light with an interval of 10 s each. Each pulse lasted for 10 s.

## Glucose measurement

For whole larvae glucose measurement, groups of ten larvae were pooled together, snap-frozen in liquid nitrogen and then stored at −80 °C until processing. After thawing on ice, 250 µl of phosphate-buffered saline (PBS) was added, and the larvae were sonicated with an ultrasonic homogenizer (Bandelin, SONOPLUS). Next, the solution was centrifuged at $13,000 \times g$ and 50 µL of the supernatant was used for glucose assay. Glucose concentration was determined using the BioVision Glucose Assay Kit (Biovision Inc.) according to the manufacturer's instructions.

30 dpf WT (average length: $11 \pm 0.7$ mm) and Tg(ins:p35) (average length: $11 \pm 1.3$ mm) animals were carefully selected based on comparable body size for blood sugar assessment. The animals were first killed with a high dose of tricaine. The gill region was manually dissected to expose the heart of the animals. Glass needle rinsed with heparin solution (anti-coagulant) was used to draw blood from the heart region. The glucose levels were measured using Accu-Chek (Roche, Lot: 10119750). A similar approach was used in adult fish.

## Small molecule treatment

We prepared 100 mM csn (M2772, Sigma) stock solution by dissolving the compound in in dimethyl sulfoxide (D8418, Sigma). Working concentration was prepared just before larvae treatment by diluting the stock solution in E3 medium (5 mM NaCl, 0.17 mM KCl, 0.33 mM $CaCl_2$, 0.33 mM $MgSO_4$, pH 7.4).

The prodrug, metronidazole (MTZ) was re-suspended to 10 mM in E3 medium containing 0.1% DMSO.

## TUNEL assay

After fixation, tissues were washed three times for 20 min with 1% PBSTx, followed by 15 min incubation in Proteinase K for permeabilization. Subsequently, the tissues were rinsed with deionized water and the TdT reaction and Click-iT Plus reaction was performed according to the manufacturer's instructions. Following incubation in the TdT reaction buffer for 10 min at 37 °C, the reaction buffer was removed, and the TdT reaction mixture was added to the samples for a 60-min incubation at 37 °C. Samples were rinsed with deionized water and washed with 3% BSA and 0,1% Triton X-100 in PBS for 5 min. They were then washed with PBS and incubated in Clik-iT Plus TUNEL reaction cocktail for 30 min at 37 °C, protected from light. Subsequently, the samples were washed with 3% BSA in PBS for 5 min, rinsed with PBS, and immunohistochemistry was performed as described above.

## Single-cell suspension of zebrafish pancreatic islets

Single-cell suspension of adult zebrafish pancreatic islet was performed according to the protocol outlined previously (Janjuha et al, 2018). For the dissociation of pancreatic islets from zebrafish

larvae (5 days post fertilization), a different protocol was followed. The pancreatic tissue of each larvae was manually dissected with tweezer forceps and dissected region was placed in 1.5 mL tube containing phosphate-buffered saline (PBS) to keep cells alive during all the dissections. Around 25–30 dissected pancreatic regions were pooled together in a 1.5 mL tube. Each sample was then enzymatically dissociated for 40–60 min at 37 °C. For this, we have developed/optimized a protocol for efficient dissociation from the early embryonic stage up to 5 dpf. A combination of 800 µL trypsin (Thermo Fischer, 15090046) and 400 µL Dispase II (Merck, SCM133) was best suited for efficient dissociation. During the dissociation, each sample was pipetted up and down every 15 min to mechanically aid in dissociation. After complete dissociation, 200 µL of fetal bovine serum (FBS) was added per 1.5 mL tube to inactivate the activity of trypsin (Addition of FBS is an additional/optional step as the low temperature in subsequent centrifugation step also inactivates enzymes). Samples were quickly spin down at $0.5 \times g$ for 10 min at 4 °C and supernatant was removed carefully without touching cell pellet. The pellet was then re-suspended in 200 µL Ca2+, Mg2+-free ice-cold HBSS (Hank's Balanced Salt Solution) and passed through a 30 µm cell strainer (CellTrics® 30 µm). In the next step, 200 µl of 4% BSA in HBSS was added to the cell strainer and allowed to pass through. Calcein; fluorescent dye, a marker of live cells, was added before flow cytometry analysis and sorting. the cell suspension passed through a fluorescent activated cell sorter (BDFACS Aria II and BDFACS Aria III) to separate Kaede+ and Calcein+ from negative fractions, which were collected in 2 µl of lysis buffer constituting of nuclease-free water, Triton X-100 and 4 U murine Rnase inhibitor (NEB). Lysis buffer rapidly lyse the cells and prevents the degradation of RNA.

## Single-cell RNA sequencing of zebrafish beta cells

The cells are sorted with an Aria III FACS into a 384 well plate pre-loaded with 0.5 µl freshly prepared lysis buffer (0.2% Tween-20 solution with 4 U/µl murine RNase inhibitor (NEB)). After the sort, the plate is shortly centrifuged, and lysed cells are stored at −80 °C.

After thawing the plate, the cells are immediately mixed with 0.5 µl of a primer mix containing 5 mM dNTP (Invitrogen), 0.5 µM dT-primer*, 2 U RNase Inhibitor (NEB), ERCC spike in a mix with a final dilution of $1:2 \times 10^7$. RNA is denatured for 3 min at 72 °C, followed by reverse transcription at 42 °C for 90 min after filling up to 1.5 µl with RT buffer mix for a final concentration of 1x superscript II buffer (Invitrogen), 1 M betaine, 5 mM DTT, 6 mM MgCl$_2$, 1 µM TSO-primer*, 2.3 U RNase Inhibitor and 25 U Superscript II. After synthesis, the reverse transcriptase is inactivated at 70 °C for 15 min. The cDNA is amplified using Kapa HiFi HotStart Readymix (Peqlab) at a final 1x concentration and 0.1 uM UP-primer under the following cycling conditions: initial denaturation at 98 °C for 3 min, 23 cycles [98 °C 20 s, 67 °C 15 s, 72 °C 6 min] and final elongation at 72 °C for 5 min. The amplified cDNA is purified using 0.6x volume of hydrophobic Sera-Mag SpeedBeads (GE Healthcare) resuspended in a buffer consisting of 10 mM Tris, 20 mM EDTA, 18.5% (w/v) PEG 8000, and 2 M sodium chloride solution and DNA is eluted in 12 ul nuclease-free water. The concentration of the samples is measured with a Tecan plate reader Infinite 200 pro in 384 well black flat bottom low volume plates (Corning) using AccuBlue Broad range chemistry (Biotium). For library preparation up to 700 pg cDNA (maximal

2 µl) is desiccated and rehydrated in 1 ul Tagmentation mix (1x TTBL buffer, 0.1 ul Tagment DNA Enzyme; from TruePrep DNA Library Prep Kit V2 for Illumina; Vazyme) and tagmented at 55 °C for 5 min. Subsequently, Illumina indices are added during PCR (72 °C 3 min, 98 °C 30 s, 13 cycles [98 °C 10 s, 63 °C 20 s, 72 °C 1 min], 72 °C 5 min) with 1x concentrated KAPA Hifi HotStart Ready Mix and 0.17 µM dual indexing primers. After PCR, libraries are quantified with AccuBlue Broad range chemistry, equimolarly pooled and size selected with 0.6x(right)/0.9x (left) volume Sera-Mag SpeedBeads. This is followed by Illumina sequencing on a Novaseq6000 (50 bp PE, S1 flowcell) aiming at an average sequencing depth of 0.6 mio reads per cell. dT-primer: C6-aminolinker-AAGCAGTGGTATCAACGCAGAGTCGAC TTT TTTTTTTTTTTTTTTTTTTTTTTTTTTVN, where N represents a random base and V any base beside thymidine; TSO-primer: AAGCAGTGGTATCAACGCAGAGTACATrGrGrG, where rG stands for ribo-guanosine; UP-primer: AAGCAGTGGTATCAACGCAGAGT.

## Analysis of single-cell RNA-seq of the zebrafish beta cells

The raw sequencing reads were trimmed with cutadapt (version 2.6), and mapped against the GRCz11 genome using gsnap (version 2019-06-10, REF). Counts per gene and cell were obtained with feature Counts (version 2.0.1, REF) and the Ensembl v95 annotation. The resulting counts table was analyzed in depth in R with the Seurat package (version 3.9) as described below.

First, the dataset was controlled for quality. We excluded cells with less than 200 detected genes, and cells with more than 40% of mitochondrial reads. After quality control, 650 beta cells (171 WT, 150 csn 24 h and 329 csn 48 h) were used for downstream analysis. Insulin is the highest expressed gene in all three samples, confirming that the sorted GFP+ cells are beta cells. Next, the dataset was log-normalized with a scaling factor of 10,000. We summarized the dataset with principal component analysis, and continued to work with the top ten components. Cells with similar gene expression profiles were grouped into clusters using the Leiden algorithm and a resolution of 0.4. We used Uniform Manifold Approximation and Projection to visualize the dataset in 2D space, and colored cells according to their assigned cluster or treatment condition. Next, we used the MAST method to determine marker genes, testing differential expression between cells of one cluster versus all other clusters, respectively. Only genes expressed in at least 25% of the cells in either group were tested, then ranked, and filtered according to a fold-change of 1.5 and an adjusted $p$ value of 0.05.

## Organ donors

This study included 13 pancreases from age- and BMI matched organ donors (4 with T1D; average disease duration 6.25 years, range 0–21 years, average age 24), two single and two double autoantibody-positive (AAb+) donors without T1D (average age 18), and five control donors (without diabetes, autoantibody-negative, average age 23).

## Ethical approval

Ethics Committee of the University of Bremen & nPOD/human tissue for research purposes. Organ donors are not identifiable and

anonymous, such approved analyses using tissue for research is covered by the NIH Exemption 4 (Regulation PHS 398).

## Statistical analysis

Statistical significance between two conditions was assessed using unpaired Student's *t*-test, while between multiple groups one-way ANOVA with Tukey's multiple comparison test was used. Analyses were performed using GraphPad Prism (GraphPad Software version 7.0) and significant *p* values are described in each relevant section. Error bars are reported as mean ± standard deviation (SD), unless otherwise stated.

# Data availability

All data, code, and materials used in the analyses are available upon request. The single-cell transcriptomic dataset of TRPV-expressing beta cells is available on Gene Expression Omnibus (GEO) under accession number GSE261729: https://www.ncbi.nlm.nih.gov/geo/query/acc.cgi?acc=GSE261729&targ=self&view=quick&form=html. The mathematical models are available at https://identifiers.org/morpheus/M2986.

The source data of this paper are collected in the following database record: biostudies:S-SCDT-10_1038-S44318-024-00332-w.

# Peer review information

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

## Acknowledgements

We are grateful to the following facilities at the CMCB and the CRTD: deep sequencing, flow cytometry, light microscopy, and zebrafish facility. We are grateful to David Prober for sharing the TRPV plasmid. Estimation of mathematical model parameters was performed on HPC resources granted by the ZIH at TU Dresden. We are grateful to Kazu Kikuchi and Victor Chang Cardiac Research Institute for sharing the *TgBAC(foxp3a:TagRFP,cryaa:EGFP)$^{vcc3}$ and foxp3a$^{vcc6}$*. This research was supported by the Network for Pancreatic Organ donors with Diabetes (nPOD; RRID:SCR_014641; https://npod.org/welcome/), a collaborative type 1 diabetes research project supported by Breakthrough T1D and The Leona M. & Harry B. Helmsley Charitable Trust. The content and views expressed are the responsibility of the authors and do not necessarily reflect the official view of nPOD. Organ Procurement Organizations (OPO) partnering with nPOD to provide research resources are listed at https://npod.org/for-partners/npod-partners/. We express our deep gratitude to the donors and their families. LB acknowledges support from the BMBF (grant 031L0293D). NN acknowledges support by DFG Research Grants (NI 1495/2-1, NI 1495/2-3, JU 3081/4-1, NI 1495/5-1, MA 4172/15-1), DFG-International Research Training Group (IRTG 2251). KM acknowledges support by the DFG Research Grant (MA 4172/15-1).

## Author contributions

**Mohammad Nadeem Akhtar**: Conceptualization; Validation; Visualization; Methodology; Writing—original draft; Writing—review and editing. **Alisa Hnatiuk**: Methodology; Writing—original draft; Writing—review and editing. **Luis Delgadillo-Silva**: Methodology. **Shirin Geravandi**: Methodology; Writing—original draft. **Katrin Sameith**: Conceptualization; Methodology; Writing—original draft. **Susanne Reinhardt**: Methodology; Writing—original draft. **Katja Bernhardt**: Data curation; Methodology. **Sumeet Pal Singh**: Visualization; Methodology. **Kathrin Maedler**: Resources; Supervision; Investigation; Visualization; Methodology; Writing—review and editing. **Lutz Brusch**: Resources; Data curation; Software; Funding acquisition; Validation; Visualization; Methodology; Writing—original draft; Project administration; Writing—review and editing. **Nikolay Ninov**: Conceptualization; Resources; Data curation; Software; Formal analysis; Supervision; Funding acquisition; Validation; Investigation; Visualization; Methodology; Writing—original draft; Project administration; Writing—review and editing.

Source data underlying figure panels in this paper may have individual authorship assigned. Where available, figure panel/source data authorship is listed in the following database record: biostudies:S-SCDT-10_1038-S44318-024-00332-w.

## Funding

## Disclosure and competing interests statement

The authors declare no competing interests.

