## [Peer Review File · The EMBO Journal]

Developmental beta-cell death orchestrates the islet's inflammatory milieu by regulating immune system crosstalk

Mohammad Akhtar, Alisa Hnatiuk, Luis Silva, Shirin Geravandi, Katrin Sameith, Susanne Reinhardt, Katja Bernhardt, Sumeet Singh, Kathrin Maedler, Lutz Brusch, and Nikolay Ninov

Corresponding author: Nikolay Ninov (nikolay.ninov@tu-dresden.de)

Review Timeline:

Submission Date:	7th Mar 24
Editorial Decision:	18th Apr 24
Revision Received:	16th Sep 24
Editorial Decision:	23rd Oct 24
Revision Received:	6th Nov 24
Accepted:	18th Nov 24

Editor: Daniel Klimmeck

Transaction Report:

Dear Dr Ninov,

Thank you for submitting your manuscript for consideration by the EMBO Journal, as well as for your patience with our feedback at this time of the year. Your work has now been seen by three referees with expertise in pancreas development and beta-cell biology and we received comments from all of them which are shown below.

Given the overall interest stated and broader angle of your findings, we are able to invite you to revise your manuscript experimentally to address the referees' comments. I need to stress though that we do require strong support from the referees on a revised version of the study in order to move on to publication of the work. Given the open outcome of the extensive revision experiments I suggest considering EMBO Reports as an alternative venue for this work.

I would appreciate if you could contact me during the next weeks for exchange e.g. a video call to discuss your perspective on the comments and potential plan for revisions.

Please feel free to also contact me if you have any questions or need further input on the referee comments.

When submitting your revised manuscript, please carefully review the instructions below.

Please feel free to approach me any time should you have additional questions related to this.

Thank you for the opportunity to consider your work for publication.

I look forward to your revision.

Kind regards,

Daniel Klimmeck

Daniel Klimmeck, PhD
Senior Editor
The EMBO Journal

Instruction for the preparation of your revised manuscript:

- 1) a .docx formatted version of the manuscript text (including legends for main figures, EV figures and tables). Please make sure that the changes are highlighted to be clearly visible.
- 2) individual production quality figure files as .eps, .tif, .jpg (one file per figure).
- 3) a .docx formatted letter INCLUDING the reviewers' reports and your detailed point-by-point response to their comments. As part of the EMBO Press transparent editorial process, the point-by-point response is part of the Review Process File (RPF), which will be published alongside your paper.
- 4) a complete author checklist, which you can download from our author guidelines ([https://wol-prod-cdn.literatumonline.com/pb-assets/embo-site/Author Checklist%20-%20EMBO%20J-1561436015657.xlsx](https://wol-prod-cdn.literatumonline.com/pb-assets/embo-site/Author%20Checklist%20-%20EMBO%20J-1561436015657.xlsx)). Please insert information in the checklist that is also reflected in the manuscript. The completed author checklist will also be part of the RPF.
- 5) Please note that all corresponding authors are required to supply an ORCID ID for their name upon submission of a revised manuscript.
- 6) It is mandatory to include a 'Data Availability' section after the Materials and Methods. Before submitting your revision, primary datasets produced in this study need to be deposited in an appropriate public database, and the accession numbers and

database listed under 'Data Availability'. Please remember to provide a reviewer password if the datasets are not yet public (see <https://www.embopress.org/page/journal/14602075/authorguide#datadeposition>).

7) Our journal encourages inclusion of *data citations in the reference list* to directly cite datasets that were re-used and obtained from public databases. Data citations in the article text are distinct from normal bibliographical citations and should directly link to the database records from which the data can be accessed. In the main text, data citations are formatted as follows: "Data ref: Smith et al, 2001" or "Data ref: NCBI Sequence Read Archive PRJNA342805, 2017". In the Reference list, data citations must be labeled with "[DATASET]". A data reference must provide the database name, accession number/identifiers and a resolvable link to the landing page from which the data can be accessed at the end of the reference. Further instructions are available at .

8) At EMBO Press we ask authors to provide source data for the main and EV figures. Our source data coordinator will contact you to discuss which figure panels we would need source data for and will also provide you with helpful tips on how to upload and organize the files.

Numerical data can be provided as individual .xls or .csv files (including a tab describing the data). For 'blots' or microscopy, uncropped images should be submitted (using a zip archive or a single pdf per main figure if multiple images need to be supplied for one panel). Additional information on source data and instruction on how to label the files are available at .

9) We replaced Supplementary Information with Expanded View (EV) Figures and Tables that are collapsible/expandable online (see examples in <https://www.embopress.org/doi/10.15252/emj.201695874>). A maximum of 5 EV Figures can be typeset. EV Figures should be cited as 'Figure EV1, Figure EV2' etc. in the text and their respective legends should be included in the main text after the legends of regular figures.

11) For data quantification: please specify the name of the statistical test used to generate error bars and P values, the number (n) of independent experiments (specify technical or biological replicates) underlying each data point and the test used to calculate p-values in each figure legend. The figure legends should contain a basic description of n, P and the test applied. Graphs must include a description of the bars and the error bars (s.d., s.e.m.).

We realize that it is difficult to revise to a specific deadline. In the interest of protecting the conceptual advance provided by the work, we recommend a revision within 3 months (17th Jul 2024). Please discuss the revision progress ahead of this time with the editor if you require more time to complete the revisions.

Referee #1:

EMBOJ-2024-117149

Title: "Developmental beta cell death orchestrates the islet's inflammatory milieu by regulating the immune system repertoire"

Comments to the Author

The authors used a genetic model of caspase inhibition in beta cells and discovered that beta cell death occurs during juvenile zebrafish growth, which regulates beta cell mass. The authors showed that macrophage colonization correlates with the onset of beta cell apoptosis during development. They showed that activation of TRPV triggered a Ca²⁺ overload, which triggered beta-cell apoptosis. Inhibition of beta-cell death leads to a reduction in the number of islet-resident proinflammatory macrophages and an increase in Treg cells. Treg deficiency caused an activation of NFκB signaling. The authors concluded that beta-cell apoptosis shifts the immune cell repertoire towards pro-inflammatory.

Major critics

Overall, the manuscript provides important new data that will be of interest to the general community. However, there are several points, that, if addressed, will further significantly strengthen the manuscript.

1. In Fig. 1. In the islets, many insulin/mCherry expressing cells are found at 5, 15, and 30 dpf. The authors claim more insulin-positive cells in the p35 Tg islets. It seems that there are more insulin-negative cells in the wild-type (WT) islets compared to the p35 transgenic line. What are these insulin-negative cells? The authors should explore the differences in the components of the endocrine islet cells in P35 overexpressing transgenic lines compared to the WT islets. Is the total cell number in the WT islets less compared to the p35 Tg islets? Do the p35 Tg islets also have more other endocrine cells than the WT islets? Are the size of the p35 Tg islets larger than the WT islets? The authors should characterize the cell component of the WT vs p35 Tg islets. Did the cells turn off their endocrine marker expressions before undergoing apoptosis? The authors should perform co-staining the islets with other endocrine markers, Pdx1, Nkx6.1, and Aldh1a3, in addition to insulin staining.

2. How many independent experiments were performed in each Figure? This should be mentioned in the figure legend. What do the n stand for? Is n stands for the total number of individual fish from several experiments? This should be clarify.

Minor critics

1. Figures should be referred to in the text in the sequence in which they appear. The authors did not refer to every Figure in the text; sometimes, the order is not in sequence. Similarly, with the supplementary figures. No text referring to Fig 4A. The text referring to Supplementary Fig 4D comes earlier than Sup. Fig 4C, but no refer to Sup Fig 4A, 4B. For example. Please check the entire manuscript again.

2. The format for reference seems not to be correct. Please check. The reference should not be in numeric order but in alphabetic order in the reference list. Anyway, it is hard to read and find the correct reference because it is not in alphabetic order.

Referee #2:

In this study, Ninov and colleagues used a combination of genetic approaches, mathematical modelling and single-cell transcriptomics to study beta-cell death and its potential role(s) in islet formation in zebrafish. Previous descriptive work in rodent islets showed a peak of cell death 2-3 weeks after birth. Here, the authors took advantage of the fish as a powerful genetic model and established various transgenic lines to block or induce cell death in vivo, specifically in beta-cells, and study the consequences. They found an expansion of the islet size upon genetic inhibition of cell death in a transgenic zebrafish model that expressed the caspase inhibitor p35 in insulin-positive cells. Based on these results, they suggested that beta-cell death plays a role in the establishment of proper beta-cell mass. Additionally, they showed that beta-cell death influences the islet's inflammatory milieu by regulating the immune system cell repertoire, attracting macrophages, and reducing T regulatory cells, with an active anti-inflammatory role.

This is an interesting study which sheds light on the underexplored question, the contribution of cell death to tissue development, and might open new avenues of investigation in the field of T1D disease mechanisms. Specific concerns with the manuscript that need to be addressed to strengthen the conclusions drawn by the authors are listed below.

Main Concerns:

1) The transgenic zebrafish model, which expresses the caspase inhibitor p35 in insulin-positive cells, should be better characterized. Specifically, the inhibition of cell death should be assessed to validate the model. Are the p35 transgenic beta-

cells more resistant to stress or ectopic induction of apoptosis, such as for example in the chemogenetic model of beta-cell excitotoxicity?

2) There is quite a lot of heterogeneity among macrophages, even just among the tissue-resident ones in distinct organs. Is the mpeg marker a universal macrophage marker in the pancreas, labelling and targeting all macrophages? Is it specific for macrophage in adult fish? Ferrero et al 2020 (doi: 10.1002/JLB.1A1119-223R) reported that in addition to macrophages a subpopulation of B-lymphocytes is marked by mpeg reporters in most adult zebrafish organs. Have the authors checked for this?

3) The study makes use of many different transgenic models, which can be an advantage, but sometimes it can complicate the interpretation of the results obtained. When using too many different models, one needs to establish how comparable they are to each other and the rationale for using them indiscriminately throughout the study. For example, are there any differences between Tg(mpeg1:EYFP-NTR) line, in which macrophages are genetically ablated, and the irf8^{-/-} mutant line? The two models are used here in different experiments, but it is unclear if their phenotypes are the same.

4) The chemogenetic model of beta-cell excitotoxicity [Tg(ins:TRPV);Tg(ins:Kaede)] raises some concerns, its characterization should be expanded, and the interpretation of some of the results could be more careful. What are the advantages of using this model compared to the well established nitroreductase-mediated beta-cell ablation system in zebrafish (Curado et al 2009), for example using the Tg(ins:CFP-NTR) line?

Also, the evidence of inflammation and changes in immune cell repertoire shown by scRNAseq in the Tg(ins:TRPV);Tg(ins:Kaede) islets are not very robust, being supported by only very few markers (see Fig 5D). These results should be validated using the NF- κ B reporter or more specific markers (e.g., FoxP3, mpeg or CD45).

5) The expression levels of pro-inflammatory cytokines, IL-1 β or TNF α , in the beta cells of wild-type animals should be shown in the Supplementary information or dropped, but this cannot be reported as data not shown.

Referee #3:

General Summary: In this manuscript by Akhtar et al., the authors aim to characterize the extent of beta cell death during the development of the islet and associated activation and shaping of the immune system. The authors develop clever new transgenic zebrafish lines to test their hypotheses, including a novel excitotoxic model of beta cell death. While interesting, the manuscript fails to deliver a new message that couldn't otherwise be gleaned from a multitude of studies in the literature—namely, that beta cell death subsequently affects the immune response. In the context of type 1 diabetes, this phenomenon is well described in several studies (e.g. PMIDs 30799288, 35767947, and many others). The context of the study focuses heavily on type 1 diabetes and autoimmunity in mammalian systems, yet neither type 1 diabetes nor mammals have been studied. Beyond this overall message, several technical concerns limit the enthusiasm of the findings.

Major Concerns:

1. The sample size for all experiments is small—6 samples or fewer. Given the relative ease of obtaining large sample sizes in zebrafish, this is quite low.
2. The developmental time window in which islet size increases 1.5X in the p35-expressing animals should be determined to a greater resolution than the 15-day window presented here. Is apoptosis (of over 100 beta cells per islet) happening chronically during this 2-week window, or is it triggered en masse at some developmental milestone (as suggested in the discussion)? The tight clustering of 'beta cells per islet' data points at both 15 and 30 dpf timepoints and the delay in the appearance of macrophage infiltration until after 30 dpf suggests that the process is stereotypical and starts within the 15-day window examined. However, it is unclear from the data included whether this remodeling is discrete in time or ongoing beyond 30 dpf.
3. PCNA staining marks cells in the G1/S phase when proliferating and some DNA repairs. Some have shown a poor correlation between PCNA and direct labeling of DNA replication with BrDU (PMID: 14623936). Given this, and that labeled uridine methods like EdU are more sensitive for detecting low proliferation rates (as labeling windows can be lengthened), the study should incorporate extended EdU-labeling at stages between 15 and 30 dpf to rule out/reveal ongoing proliferation at low levels that could be missed by snapshot labeling with small sample sizes.
4. There is no difference in the beta cell number at 5 dpf, yet this is the time point chosen to examine beta cell functionality (GcAMP and free glucose assay). Because the study shows no significant changes in beta cell number between wt and the p35-expressing fish, this is not altogether surprising. These assays should be done in older fry before, during, and after the observed culling of beta cells. With regard to the glucose assays, the manuscript should indicate if the free glucose data come from one experiment with triplicate pools of 10 embryos or three separate experiments, 1 pool each.
5. When the authors propose that only a small fraction of the beta cells are proliferative (figure 1F), are they commenting on heterogeneity in the beta cell mass (as published in PMID: 19706417)? Or rather, are they suggesting that at a given snapshot in time, only a fraction of replication-competent beta cells is proliferating, i.e., most beta cells have a long quiescent refractory period? Would these alternate cases impact the mathematical model, and can the authors discuss this? Furthermore, the Hesselson/Stainier Study should be included in this context.
6. The authors should determine other overt changes in zebrafish body habitus or behavior that can be observed in the p35-expressing fish, such as activity level, metabolism, feeding behavior, body mass, and the quantity and character of other islet cell types.

7. *Macrophages appear rapidly in 3-6 hours in response to beta cell distress in the ins:NTR model (reported in PMID: 29785241). This is consistent with the authors' assertion that apoptotic beta cells are difficult to detect and quantify because of the swift immune response and should be included in the associated discussions. However, macrophage accumulation in the WT islet was examined at 33, 15, and 5 dpf in otherwise WT islets, and at 33 dpf is the first appearance of islet macrophages. This does not seem to fit the timeline established earlier in the manuscript, where islet size was already 1.5X larger by 30 dpf. If beta cells had already been under apoptosis and been ingested by macrophages between 15 and 30 days in WT animals, would macrophages not be apparent earlier? Again, this underscores the need for a more precise determination of the time of onset of beta cell number increase in the p35 fish and the time of onset of macrophage islet infiltration in WT fish.

8. While the authors show depletion of beta cell quantity in the TRPV/csn model, the authors need to show whether there is a change in the free glucose of treated larvae. Systemic levels of glucose may impact the proliferation or neo-differentiation of beta cells. Furthermore, since kaede conversion and edu labeling were performed simultaneously, can the authors comment on whether there was any difference notable in proliferation between the green only (new, younger) and the yellow (pre-existing, older) beta cells-i.e., are the neogenic beta cells that have presumably lower expression of the TRPV (as it is ramping up based on insulin promoter activity in newly differentiated beta cells) and perhaps less activation of Ca²⁺ flux proliferating at a different rate?

9. What is the status of the islet size in the fish examined at 6 mos.? What is the free glucose/function of the beta cell complement? These experiments should accompany the analysis of immune infiltration and expression of NFkB.

10. The discussion appears to overinterpret the data-much is made of mammalian relevance, yet no data from mammalian model systems is presented in this study. While it is laudable that the authors are attempting to relate their findings to mammalian systems, the data frankly provide no evidence of a link, and this aspect needs to be toned down considerably.

11. The title over-dramatizes the findings; for example, it includes the term "immune repertoire", but the authors do not actually study a repertoire of immune cells (only macrophages and Tregs). Additionally, the term "orchestrates" is a stretch, as no evidence is provided for larger-scale governance or "regulation" per se of the immune system by beta cell death. Finally, the title omits any reference to zebrafish, which is the entire focus of this study and, therefore, is misleading. A more concise and accurate title should be considered (e.g. Developmental beta cell death governs the islet inflammatory milieu in zebrafish).

Minor Concerns:

Check the manuscript carefully for typographical errors.

Additional:

A germane question that the authors are positioned to answer with their tools is whether the p35 expression can rescue cell death in TRPV/can-stressed cells.

Referee #1:

EMBOJ-2024-117149

Title: "Developmental beta cell death orchestrates the islet's inflammatory milieu by regulating the immune system repertoire"

We are grateful to all reviewers for providing us with constructive and supportive feedback. We have addressed the major points and all other, more minor concerns. The major revisions include:

1. We performed a quantification of beta-cell number and macrophage colonization at a higher time resolution, as well as EdU analysis of proliferation. These results strengthened our conclusions that cell death during development plays a critical role in shaping beta-cell mass and coordinates the crosstalk and the islet colonization by macrophages.
2. The new quantifications of beta cell numbers were used to improve the mathematical model and to distinguish between a gradual cell death or a sudden event in which many cells die in a short period. We have also confirmed the identified mechanism of heterogeneous proliferative subpopulations and chronic apoptosis using a stochastic cell-based model, implemented in the software Morpheus (Figure 2).
3. We carried out a histological validation of our RNA sequencing results using reporters of inflammation and histochemistry in combination with caspase inhibition. Taken together with our transcriptomic analysis, these results reveal that metabolically-stressed beta cells follow paths towards apoptosis or dedifferentiation with caspase inhibition able to rescue the cells from apoptosis but not from dedifferentiation.
4. Our colleague, Dr. Kathrin Maedler (Univ. of Bremen), an expert on human tissue analysis in subjects with Type 1 Diabetes, has contributed data showing an increase in macrophages within islets of organ donors positive for one or two autoantibodies but not yet diagnosed with T1D. These individuals also present a slight but significant reduction in beta cell mass. Since these data increase the relevance of our work to T1D, we included them in the manuscript (Supp. Figure 4).
5. We have also tackled all other points in terms of improving data analysis, technical aspects, and readability. For example, we provided additional measurements of glucose levels, analyzed other endocrine cells in the p35 model, and complemented the mpeg1-macrophage population analysis with a new transgenic reporter line that we have generated based on the recently published mfap4 promoter.

Comments to the Author

The authors used a genetic model of caspase inhibition in beta cells and discovered that beta cell death occurs during juvenile zebrafish growth, which regulates beta cell mass. The authors showed that macrophage colonization correlates with the onset of beta cell apoptosis during development. They showed that activation of TRPV triggered a Ca²⁺ overload, which triggered beta-cell apoptosis. Inhibition of beta-cell death leads to a reduction in the number of islet-resident proinflammatory macrophages and an increase in Treg cells. Treg deficiency caused an activation of NFκB signaling. The authors concluded that beta-cell apoptosis shifts the immune cell repertoire towards pro-inflammatory.

Major critics

Overall, the manuscript provides important new data that will be of interest to the general community. However, there are several points, that, if addressed, will further significantly strengthen the manuscript.

Response: We would like to thank the reviewer for taking the time to provide us with insightful and constructive comments, which were invaluable in improving the manuscript. We are happy that the

reviewer finds our work to be important and that it will be of interest to the general community. We have performed additional experiments and data analysis to address the concerns and improve the study.

1. In Fig. 1. In the islets, many insulin/mCherry expressing cells are found at 5, 15, and 30 dpf. The authors claim more insulin-positive cells in the p35 Tg islets. It seems that there are more insulin-negative cells in the wild-type (WT) islets compared to the p35 transgenic line. What are these insulin-negative cells? The authors should explore the differences in the components of the endocrine islet cells in P35 overexpressing transgenic lines compared to the WT islets. Is the total cell number in the WT islets less compared to the p35 Tg islets? Do the p35 Tg islets also have more other endocrine cells than the WT islets? Are the size of the p35 Tg islets larger than the WT islets? The authors should characterize the cell component of the WT vs p35 Tg islets.

Response: The reviewer asks whether the inhibition of cell death in beta-cells leads to a change in the numbers of other endocrine cells and the area of the islets. To address this question, we analyzed both the whole endocrine compartment and specific cell types. Interestingly, we found that the increase in beta-cell number is accompanied by an adaptation in which the number of alpha cells also scales up (**Fig. R1**). This suggests that a crosstalk mechanism ensures that the overall proportion of endocrine cells remains constant when beta-cells are increased. We would like to investigate this interesting behavior in the future. The counter-regulatory role of alpha cells may ensure that the overall glucose levels remain similar with controls despite having more beta-cells (**supp. figure 2 E**). In terms of the size of the islet, we measured beta-cell area and found that it is indeed increased (**supp. fig. 2F**).

Figure for reviewers removed

Did the cells turn off their endocrine marker expressions before undergoing apoptosis? The authors should perform co-staining the islets with other endocrine markers, Pdx1, Nkx6.1, and Aldh1a3, in addition to insulin staining.

Response: Thank you for this suggestion. Indeed, our RNA seq of the TRPV model suggest that cells in cluster 4 show activation of inflammation and cell death genes (**Fig. 5D**). In contrast to cluster 4, beta cells in cluster 5 did not show the upregulation of cell death and immune-related genes. Instead, they showed a more prominent downregulation of bona-fide beta cell markers including *ins* and *pdx1*, and upregulated expression of the canonical marker of mammalian beta-cell dedifferentiation *aldh1a3* (**Fig. 5D**). To test whether the dying cells lose endocrine markers at the histological level, we performed Pdx1 antibody staining after the induction of the TRPV model, as this is the only working antibody-marker that is available for zebrafish beta-cells. We observed an increased proportion of Pdx1-low beta-cell compared to controls (**Supp. Fig. 8D-F**). These cells tend to exhibit low insulin expression, consistent with the reduced of *ins* and *pdx1* expression in our RNA seq data. Subsequently, we used the photoconversion approach to distinguish pre-existing from newly born beta-cells and then induced the TRPV model. We found that pre-existing beta-cells turn down- Pdx1 expression in response to stress (**Supp. Fig. 9A,B**). We then ablated the macrophages in order to reveal beta-cell corpses and observe the levels of Pdx1. The dying cells, which showed condensed and pycnotic nuclei, had lost their nuclear

Pdx1 expression (**Supp. Fig. 9C**) (n = 13 cells from 6 larvae). Interestingly, while forcing p35-expression in the TRPV model partially rescued the decline in beta-cell number (**Supp. Fig. 9D-F**), it was unable to prevent Pdx1-downregulation (**Supp. Fig. 9A,B**). This suggests that caspase inhibition can rescue the cells from apoptosis but not from dedifferentiation. Taken together with our transcriptomic analysis, these results reveal that metabolically-stressed beta cells follow paths towards apoptosis or dedifferentiation.

This text is added in lines 326-340.

2. How many independent experiments were performed in each Figure? This should be mentioned in the figure legend. What do the n stand for? Is n stands for the total number of individual fish from several experiments? This should be clarify.

Response: We have now indicated the number of independent samples for each experiment in each figure legend.

Minor critics

1. Figures should be referred to in the text in the sequence in which they appear. The authors did not refer to every Figure in the text; sometimes, the order is not in sequence. Similarly, with the supplementary figures. No text referring to Fig 4A. The text referring to Supplementary Fig 4D comes earlier than Sup. Fig 4C, but no refer to Sup Fig 4A, 4B. For example. Please check the entire manuscript again.

Response: Thank you for the comment. We rechecked all labels and now they seem to be in order.

2. The format for reference seems not to be correct. Please check. The reference should not be in numeric order but in alphabetic order in the reference list. Anyway, it is hard to read and find the correct reference because it is not in alphabetic order.

Response: We made sure that they are in an alphabetic order.

Referee #2:

In this study, Ninov and colleagues used a combination of genetic approaches, mathematical modelling and single-cell transcriptomics to study beta-cell death and its potential role(s) in islet formation in zebrafish. Previous descriptive work in rodent islets showed a peak of cell death 2-3 weeks after birth. Here, the authors took advantage of the fish as a powerful genetic model and established various transgenic lines to block or induce cell death in vivo, specifically in beta-cells, and study the consequences. They found an expansion of the islet size upon genetic inhibition of cell death in a transgenic zebrafish model that expressed the caspase inhibitor p35 in insulin-positive cells. Based on these results, they suggested that beta-cell death plays a role in the establishment of proper beta-cell mass. Additionally, they showed that beta-cell death influences the islet's inflammatory milieu by regulating the immune system cell repertoire, attracting macrophages, and reducing T regulatory cells, with an active anti-inflammatory role.

This is an interesting study which sheds light on the underexplored question, the contribution of cell death to tissue development, and might open new avenues of investigation in the field of T1D disease mechanisms. Specific concerns with the manuscript that need to be addressed to strengthen the conclusions drawn by the authors are listed below.

Response: We would like to thank the reviewer for taking the time to provide us with insightful and constructive comments, which were invaluable in improving the manuscript. We are happy that the reviewer finds our work interesting and exploring new questions that are relevant to T1D. We have performed additional experiments and data analysis to address the concerns and improve the study.

Main Concerns:

1) The transgenic zebrafish model, which expresses the caspase inhibitor p35 in insulin-positive cells, should be better characterized. Specifically, the inhibition of cell death should be assessed to validate the model. Are the p35 transgenic beta-cells more resistant to stress or ectopic induction of apoptosis, such as for example in the chemogenetic model of beta-cell excitotoxicity?

Response: Thank you for this suggestion. To address whether the p35 transgenic beta-cells are more resistant to induction of apoptosis, we forced p35-expression in the TRPV model. TRPV induction leads to cell death of a subset of beta-cells that turn on inflammation and cell death pathways (as seen in cluster 4 in our RNA seq analysis). Forcing p35-expression in the TRPV model rescued partially the overall decline in beta-cell numbers (**Supp. Fig. 9D-F**). We hypothesized that p35 does not fully block the loss of cells in the TRPV model as it might specifically block apoptosis but not dedifferentiation (loss of cell identity). To test this idea, we first performed Pdx1 antibody staining after induction of cell stress using the TRPV model. We observed an increased proportion of Pdx1-low beta-cell compared to controls (**Supp. Fig. 8D-F**). These cells tend to exhibit low insulin expression, consistent with the reduced of *ins* and *pdx1* expression in our RNA seq data. Subsequently, we used the photoconversion approach to distinguish pre-existing from newly born beta-cells and then induced the TRPV model (**Supp. Fig. 9A,B**). We found that pre-existing beta-cells turn down- Pdx1 expression in response to stress. Forcing p35-expression in the TRPV model did not prevent Pdx1-downregulation in beta-cells (**Supp. Fig. 9A,B**), suggesting that it does not rescue the loss of cell identity. In summary, p35 expression can protect the beta-cells from stress-induced cell death but does not block dedifferentiation.

This text is added in lines: 326-340.

2) There is quite a lot of heterogeneity among macrophages, even just among the tissue-resident ones in distinct organs. Is the mpeg marker a universal macrophage marker in the pancreas, labelling and targeting all macrophages? Is it specific for macrophage in adult fish ? Ferrero et al 2020 (doi: 10.1002/JLB.1A1119-223R) reported that in addition to macrophages a subpopulation of B-lymphocytes is marked by mpeg reporters in most adult zebrafish organs. Have the authors checked for this?

Response: Since the *mpeg1* promoter has been shown to also mark B-cells in zebrafish by Valérie Wittamer's group ¹, we generated a line expressing mTurquoise under the macrophage *mfap4* promoter established by the Tobin's group ² and found that the majority of *mpeg1*-positive cells in the islets are also *mfap4*-positive and exhibit a reduction in the islets of p35 fish (**Supp. Fig. 10**), thereby confirming our conclusions.

3) The study makes use of many different transgenic models, which can be an advantage, but sometimes it can complicate the interpretation of the results obtained. When using too many different models, one need to establish how comparable they are to each other and the rationale for using them indiscriminately throughout the study. For example, are there any differences between Tg(*mpeg1*:EYFP-NTR) line, in which macrophages are genetically ablated, and the *irf8*^{-/-} mutant line? The two models are used here in different experiments, but it is unclear if their phenotypes are the same.

Response: Indeed, each model has different advantages and disadvantages. Tg(*mpeg1*:EYFP-NTR) provides very good ablation efficiency in the larvae and it is more convenient to use compared to IRF8 mutants, as there is no need for extensive genotyping of homozygous fish in the background of the TRPV model. The two transgenic lines can simply be crossed and the double-transgenic progeny analyzed visually. However, the Tg(*mpeg1*:EYFP-NTR) does not provide a robust ablation at later stages and macrophages quickly recover with high variability across individual fish. Hence, for the later stages of our analysis, we used the less convenient but more consistent IRF8 mutant to reduce the macrophage number in juveniles.

4) The chemogenetic model of beta-cell excitotoxicity [Tg(ins:TRPV);Tg(ins:Kaede)] raises some concerns, its characterization should be expanded, and the interpretation of some of the results could be more careful. What are the advantages of using this model compared to the well established nitroreductase-mediated beta-cell ablation system in zebrafish (Curado et al 2009), for example using the Tg(ins:CFP-NTR) line?

Response: The Tg(ins:CFP-NTR) provides a very robust model of cell ablation. It works by converting MTZ into a toxic product. This toxic product directly intercalates and crosslink the DNA in the cells³. Hence, it is a model of cell ablation that induces an unrepairable, severe and complete damage to the cells that does not reflect a more physiological process in which cell exhaustion leads to downstream events linked to cell death. Typically, beta-cells experience oscillations in Ca²⁺ to release insulin. We hypothesized that by enhancing the Ca²⁺ levels using the TRPV model, we will be able to induce a more physiological model of beta-cell stress. Indeed, Ca²⁺ excitotoxicity is one of the principal components of metabolic stress in beta cells (reviewed in^{4,5}). In addition, we now show that the TRPV fish experience hyperglycemia after induction (**Fig. 4G**). Hence, the TRPV model provides a more physiological setup to induce beta-cell stress and eventually cell death as compared to the previous NTR system. We have now made this point in the text, explaining the rationale behind the need for a new system that induces beta-cell stress without directly killing them. Lines 215-221.

Also, the evidence of inflammation and changes in immune cell repertoire shown by scRNAseq in the Tg(ins:TRPV);Tg(ins:Kaede) islets are not very robust, being supported by only very few markers (see Fig 5D). These results should be validated using the NF-kB reporter or more specific markers (e.g., FoxP3, mpeg or CD45).

Response: To validate our results at the protein level, we took advantage of the Tg(NF-kB:EGFP) reporter line in which the GFP is expressed under the control of six tandem NF-kB binding region. We crossed Tg(NF-kB:EGFP) to Tg(ins:TRPV) and monitored beta-cell inflammation after 48h of csn treatment. We detected an activation of GFP expression in 10 % of beta-cells (**Supp. Fig. 8A,B**), which is consistent with the proportion of cells seen in cluster 4 (cells expressing cell death and inflammation markers). As indicated above, we also validated the data in terms of the process of de-differentiation by photo-conversion experiments and immunohistochemistry for Pdx1 and Insulin. Lines 320-326.

5) The expression levels of pro-inflammatory cytokines, IL-1 β or TNF α , in the beta cells of wild-type animals should be shown in the Supplementary information or dropped, but this cannot be reported as data not shown.

Response: We have now provided the data as a supplementary figure (**Supp. Fig. 11**).

Referee #3:

General Summary: In this manuscript by Akhtar et al., the authors aim to characterize the extent of beta cell death during the development of the islet and associated activation and shaping of the immune system. The authors develop clever new transgenic zebrafish lines to test their hypotheses, including a novel excitotoxic model of beta cell death. While interesting, the manuscript fails to deliver a new message that couldn't otherwise be gleaned from a multitude of studies in the literature-namely, that beta cell death subsequently affects the immune response. In the context of type 1 diabetes, this phenomenon is well described in several studies (e.g. PMIDs 30799288, 35767947, and many others). The context of the study focuses heavily on type 1 diabetes and autoimmunity in mammalian systems, yet neither type 1 diabetes nor mammals have been studied. Beyond this overall message, several technical concerns limit the enthusiasm of the findings.

Response: We would like to thank the reviewer for taking the time to provide us with insightful and constructive comments, which were invaluable in improving the manuscript. We are happy that the reviewer finds our work interesting. Our study is the first systematic analysis of naturally occurring developmental beta-cell death and its impact on the immune system, which has not been addressed, as previous studies have mainly used model of beta-cell damage in adult mice. The point about the relevance to mammalian T1D is also well taken. To improve the outlook of our study for type 1 diabetes, we have added data from donors with type 1 diabetes. In addition, we did our best to improve on the technical aspects that the reviewer mentioned. Addressing the comments have improved our manuscript and we are grateful to the reviewer for the constructive criticism.

Major Concerns:

1. The sample size for all experiments is small-6 samples or fewer. Given the relative ease of obtaining large sample sizes in zebrafish, this is quite low.

Response: As suggested by the reviewer, we have now increased the time-resolution of both the macrophage colonization (**Fig. 3B**) and the beta-cell counts (**Supp. Fig. 1A,B**). In addition, we have also performed EdU analysis of beta-cell proliferation (**Supp. Fig. 1C**). This has led to more robust results that substantiate the main findings. However, we cannot provide more sample numbers for each other experiments as we are limited by the 3R principles, which apply for fish older than 5 dpf. In fact, to perform the new EdU experiments, we wrote a new animal license application that had to undergo the stringent approval process by our authorities, as these experiments involve later stage animals.

2. The developmental time window in which islet size increases 1.5X in the p35-expressing animals should be determined to a greater resolution than the 15-day window presented here. Is apoptosis (of over 100 beta cells per islet) happening chronically during this 2-week window, or is it triggered en masse at some developmental milestone (as suggested in the discussion)? The tight clustering of 'beta cells per islet' data points at both 15 and 30 dpf timepoints and the delay in the appearance of macrophage infiltration until after 30 dpf suggests that the process is stereotypical and starts within the 15-day window examined. However, it is unclear from the data included whether this remodeling is discrete in time or ongoing beyond 30 dpf.

Response: We thank the reviewer for this suggestion. The new analysis of intermediate developmental stages showed that the excess beta-cells first appear by 20 dpf in Tg(ins:p35) (**Supp. Fig. 1A,B**). These new results have been incorporated in the mathematical model (**new Fig. 2**). Please see the new exploration of 4 alternative model variants under point 5 below. These 4 models, termed A-D, consider the combinations of population homo-/heterogeneity and chronic/acute apoptosis. We find that the acute apoptosis models C and D can match the first data point with differences between WT and p35 experiments (at 20 dpf) but do not match the shallower slope of the following WT data points. We can therefore reject the hypothesis of an acute wave of apoptosis of (as tested here) just 1 day duration. The shallower slope of the WT data points between 20 and 30 dpf, as compared to the p35 time course, rather suggests chronic apoptosis at a rate of 3.0 ± 0.6 %/d.

3. PCNA staining marks cells in the G1/S phase when proliferating and some DNA repairs. Some have shown a poor correlation between PCNA and direct labeling of DNA replication with BrDU (PMID: 14623936). Given this, and that labeled uridine methods like EdU are more sensitive for detecting low proliferation rates (as labeling windows can be lengthened), the study should incorporate extended EdU-labeling at stages between 15 and 30 dpf to rule out/reveal ongoing proliferation at low levels that could be missed by snapshot labeling with small sample sizes.

Response: EdU has been performed using 24h incubations at 23 and 30 dpf (**Supp. Fig. 1C**). There were no differences in proliferation between control and p35 fish, consistent with the previous PCNA analysis.

4. There is no difference in the beta cell number at 5 dpf, yet this is the time point chosen to examine beta cell functionality (GCaMP and free glucose assay). Because the study shows no significant changes in beta cell number between wt and the p35 expressing fish, this is not altogether surprising. These assays should be done in older fry before, during, and after the observed culling of beta cells. With regard to the glucose assays, the manuscript should indicate if the free glucose data come from one experiment with triplicate pools of 10 embryos or three separate experiments, 1 pool each.

Response: Glucose measurement were previously performed at 5 dpf and at 6 months, showing no differences between controls and WT (**Supp. Fig. 2C,E**). The 5 dpf stage was chosen as a time-point as it allows to challenge the larvae with glucose and in this way, measure the functional capabilities of the beta-cells. There were no differences in glucose. The GCaMP analysis was also done at 5 dpf because it allows to perform live imaging of glucose-stimulated calcium influx, which is unfeasible at later stages. These experiments show that beta-cell functionality is not affected. To address the reviewer's concern, we have now measured the glucose at 30 dpf, as the reviewer suggested and found no differences (**Supp. Fig. 2D**). Overall, these experiments show that beta-cell function is preserved.

The information about group size in terms of glucose measurements has also now been added in the figure legend and the methods. Thanks for making this point.

5. When the authors propose that only a small fraction of the beta cells are proliferative (figure 1F), are they commenting on heterogeneity in the beta cell mass (as published in PMID: 19706417)? Or rather, are they suggesting that at a given snapshot in time, only a fraction of replication-competent beta cells is proliferating, i.e., most beta cells have a long quiescent refractory period? Would these alternate cases impact the mathematical model, and can the authors discuss this? Furthermore, the Hesselson/Stainier Study should be included in this context.

Response: We thank the reviewer for suggesting an alternative model. We have implemented this suggestion and estimated the (potentially long) cell cycle duration. As this model comparison may be influenced by the choice of incorporating a chronic or an acute apoptosis process in the model, we have extended this comparison to 4 model variants with all combinations of the above assumptions on population homo-/heterogeneity and chronic/acute apoptosis. For each model variant we have estimated all model parameters (given as median point estimate and 95% confidence interval) by minimizing the sum of weighted squared errors over the 66 individual experimental measurements with weights given by the squared mean of each experimental condition. The results are summarized in the following table R1, where Model A is best with less than half the error than the worst Model D.

Table R1. Results of parameter estimation for four variants (A-D) of the mathematical model ordered by descending quality of fit.

	Model A	Model B	Model C	Model D
Assumptions	heterogeneous, chronic apoptosis	homogeneous, chronic apoptosis	heterogeneous, acute apoptosis	homogeneous, acute apoptosis
Sum of weighted squared errors (lowest is best)	0.138	0.143	0.328	0.329
T _c in d (cell cycle)	1.05±0.09	14.0±2.4	1.7±0.2	18.5±4.4

	Model A	Model B	Model C	Model D
duration)				
v in 1/d (neogenesis)	3.3 ± 0.5	3.1 ± 0.6	4.5 ± 0.8	4.4 ± 0.8
δ in %/d (apoptosis)	3.0 ± 0.6	3.4 ± 1.3	35 ± 8	35 ± 30
T_{onset} in dpf (onset of apoptosis)	16.3 ± 1.6	17.5 ± 2.4	19.6 ± 1.2	19.6 ± 4.2
T_{duration} in d (duration of apoptosis)	estimated >14d	estimated >14d	set to 1	set to 1
$k_{\text{activation}}$ in 1/d (entry into cell cycle)	0.03 ± 0.05	n.a.	0.06 ± 0.04	n.a.
$k_{\text{inactivation}}$ in 1/d (entry into quiescence)	set to 1	n.a.	set to 1	n.a.
initial N_p in 1 (cell number at $t=5$ dpf)	set to 3	set to 44	set to 3	set to 44
initial N_q in 1 (cell number at $t=5$ dpf)	set to 41	n.a.	set to 41	n.a.

The model predictions following from the above parameter fits are compared to the experimental data in **figure R2**. Interestingly, the acute apoptosis models C and D can match the first data point with differences between WT and p35 experiments (at 20dpf) but do not match the shallower slope of the following WT data points. We therefore reject the assumption of an acute wave of apoptosis of just 1 day duration.

Figure R2. Results of model simulations (lines) for four variants (A-D) of the mathematical model using best-fit parameter values for each variant as given in table R1. Data points represent means with 2*std from experimental cell counts.

Model variants with a uniform population of beta cells (Models B and D) predict an unreasonably slow cell cycle of 14 days and longer (see **table R1**). Moreover, they predict too high PCNA+ levels as shown in **figure R3**. We therefore conclude that the observed data is best explained by Model A of a heterogeneous islet with two subpopulations, one cycling with a cell cycle duration close to 1d and a quiescent subpopulation to which the majority of cells belong.

Figure R3. Results of model simulations (lines) for four variants (A-D) of the mathematical model using best-fit parameter values for each variant as given in table R1. Percentages of cycling cells per total cell number (N_p/N) were computed from model simulations. Data points represent PCNA+ percentages from experiments as shown in main figure 1F.

We have confirmed that the identified mechanism of heterogeneous subpopulations and chronic apoptosis reproduces the above results in a stochastic cell-based model, implemented in the software Morpheus with parameter estimator FitMultiCell^{6,7} (**Figure R4** shows the simulation with cycling cells in green and the time course of total cell numbers covers the time span 5dpf-30dpf)

Figure R4. Results of model simulations for a stochastic cell-based model with the mechanisms and parameter values of model variant A from table R1. The panels show the simulation state for WT (left) and p35 (middle) with cycling cells in green, quiescent cells in red and dying cells in blue. The panel on the right shows the time courses of total cell numbers in this stochastic model. One model simulation is shown for WT and one for p35.

Finally, we have referred to the study by Hesselson *et al.* in the part of the text related to proliferation dynamics.

6. The authors should determine other overt changes in zebrafish body habitus or behavior that can be observed in the p35-expressing fish, such as activity level, metabolism, feeding behavior, body mass, and the quantity and character of other islet cell types.

Response: We have not observed any changes in behavior, glucose values and size. However, as suggested also by Rev1, we analyzed both the whole endocrine compartment and specific cell types at 30 dpf. Interestingly, we found that the increase in beta-cell number is accompanied by an adaptation in which the number of alpha cells also scales up (Fig R1). This suggests that a crosstalk mechanism ensures that the overall proportion of endocrine cells remains constant when beta-cells are increased. We would like to investigate this interesting behavior in the future. The counter-regulatory role of alpha cells may ensure that the overall glucose levels remain similar with controls despite having more beta-cells.

7. *Macrophages appear rapidly in 3-6 hours in response to beta cell distress in the ins:NTR model (reported in PMID: 29785241). This is consistent with the authors' assertion that apoptotic beta cells are difficult to detect and quantify because of the swift immune response and should be included in the associated discussions. However, macrophage accumulation in the WT islet was examined at 33, 15, and 5 dpf in otherwise WT islets, and at 33 dpf is the first appearance of islet macrophages. This does not seem to fit the timeline established earlier in the manuscript, where islet size was already 1.5X larger by 30 dpf. If beta cells had already been under apoptosis and been ingested by macrophages between 15 and 30 days in WT animals, would macrophages not be apparent earlier? Again, this underscores the need for a more precise determination of the time of onset of beta cell number increase in the p35 fish and the time of onset of macrophage islet infiltration in WT fish.

Response: We have improved the temporal resolution by including more stages and carefully scoring macrophage presence in contract with the islet by examining the islet perimeter using glucagon staining. The extended data are presented (**Figure 2A,B**). They show that indeed, the anticipated onset of beta-cell death at around 20 dpf closely matches the increase in macrophages in contract with the islet at that stage. We now indicated that this is consistent with the observations in the NTR model in the 1st paragraph of the discussion⁸. (Lines 429-431)

8. While the authors show depletion of beta cell quantity in the TRPV/csn model, the authors need to show whether there is a change in the free glucose of treated larvae. Systemic levels of glucose may impact the proliferation or neo-differentiation of beta cells.

Response: We have conducted glucose measurements after the TRPV induction and show that the larvae become hyperglycemic (**Fig. 4G**). The photo-labeling experiments indicate that there was no statistical change in neogenesis (**Fig. Suppl. Fig. 6C**).

Furthermore, since kaede conversion and edu labeling were performed simultaneously, can the authors comment on whether there was any difference notable in proliferation between the green only (new, younger) and the yellow (pre-existing, older) beta cells-i.e., are the neogenic beta cells that have presumably lower expression of the TRPV (as it is ramping up based on insulin promoter activity in newly differentiated beta cells) and perhaps less activation of Ca²⁺ flux proliferating at a different rate?

Response: We look at the data but found no statistical differences in the neogenic and pre-existing cell-proliferation between controls and p35 fish.

9. What is the status of the islet size in the fish examined at 6 mos.? What is the free glucose/function of the beta cell complement? These experiments should accompany the analysis of immune infiltration and expression of NFkB.

Response: We quantified the islet size and found a significant increase at 6 months (**Suppl. Fig. 2F**). The fasting and postprandial glucose levels at that stage were not affected (**Suppl. Fig. 2E**).

10. The discussion appears to overinterpret the data-much is made of mammalian relevance, yet no data from mammalian model systems is presented in this study. While it is laudable that the authors are attempting to relate their findings to mammalian systems, the data frankly provide no evidence of a link, and this aspect needs to be toned down considerably.

Response: To improve the link between our study and the type 1 diabetes disease, we have now reanalyzed histological data previously obtained from human donors with type 1 diabetes from nPOD⁹. We observed an increase in macrophages in the islets from organ donors positive for one or two autoantibodies; donors were not yet diagnosed with T1D (**Suppl. Fig. 4**). Since these individuals present already a slight reduction in beta cell mass⁹, it is possible that similarly to zebrafish, human macrophages recruit to the islets at the earliest instances of beta-cell damage. This text is included in lines 205-213.

11. The title over-dramatizes the findings; for example, it includes the term "immune repertoire", but the authors do not actually study a repertoire of immune cells (only macrophages and Tregs). Additionally, the term "orchestrates" is a stretch, as no evidence is provided for larger-scale governance or "regulation" per se of the immune system by beta cell death. Finally, the title omits any reference to zebrafish, which is the entire focus of this study and, therefore, is misleading. A more concise and accurate title should be considered (e.g. Developmental beta cell death governs the islet inflammatory milieu in zebrafish).

Response: We thank the reviewers for the suggestions to improve on the title of the manuscript. We agree with the reviewer that the term “repertoire” can have a broader connotation in the immunological field. Therefore, we replaced it with “crosstalk”. To keep the title concise, we prefer to exclude zebrafish in it, as the model organism is clearly stated in the abstract. We hope the reviewer will find this compromise acceptable.

Minor Concerns:

Check the manuscript carefully for typographical errors.

Response: We checked the manuscript carefully using Grammarly to fix all typographical errors.

Additional:

A germane question that the authors are positioned to answer with their tools is whether the p35 expression can rescue cell death in TRPV/can-stressed cells.

Response: To address whether the p35 transgenic beta-cells are more resistant to induction of apoptosis, we forced p35-expression in the TRPV model. TRPV induction leads to cell death of a subset of beta-cells that turn on inflammation and cell death pathways (as seen in cluster 4 in our RNA seq analysis). Forcing p35-expression in the TRPV model rescued partially the overall decline in beta-cell numbers (**Supp. Fig. 9D-F**). We hypothesized that p35 does not fully block the loss of cells in the TRPV model as it might specifically block apoptosis but not dedifferentiation (loss of cell identity). To test this idea, we first performed Pdx1 antibody staining after induction of cell stress using the TRPV model. We observed an increased proportion of Pdx1-low beta-cell compared to controls (**Supp. Fig. 8D-F**). These cells tend to exhibit low insulin expression, consistent with the reduced of *ins* and *pdx1* expression in our RNA seq. Subsequently, we used the photoconversion approach to distinguish pre-existing from newly born beta-cells and then induced the TRPV model (**Supp. Fig. 9A,B**). We found that pre-existing beta-cells turn down- Pdx1 expression in response to stress. Forcing p35-expression in the TRPV model did not prevent Pdx1-downregulation in beta-cells (**Supp. Fig. 9A,B**), suggesting that it does not rescue the loss of cell identity. In summary, p35 expression can protect the beta-cells from stress-induced cell death but does not block dedifferentiation.

Lines: 326-340.

- 1 Ferrero, G. *et al.* The macrophage-expressed gene (*mpeg*) 1 identifies a subpopulation of B cells in the adult zebrafish. *J Leukoc Biol* **107**, 431-443, doi:10.1002/JLB.1A1119-223R (2020).
- 2 Walton, E. M., Cronan, M. R., Beerman, R. W. & Tobin, D. M. The Macrophage-Specific Promoter *mfap4* Allows Live, Long-Term Analysis of Macrophage Behavior during Mycobacterial Infection in Zebrafish. *PLoS One* **10**, e0138949, doi:10.1371/journal.pone.0138949 (2015).
- 3 Edwards, D. I. Nitroimidazole drugs--action and resistance mechanisms. II. Mechanisms of resistance. *J Antimicrob Chemother* **31**, 201-210, doi:10.1093/jac/31.2.201 (1993).
- 4 Magnuson, M. A. & Osipovich, A. B. Ca(2+) signaling and metabolic stress-induced pancreatic beta-cell failure. *Front Endocrinol (Lausanne)* **15**, 1412411, doi:10.3389/fendo.2024.1412411 (2024).
- 5 Klec, C., Ziomek, G., Pichler, M., Malli, R. & Graier, W. F. Calcium Signaling in ss-cell Physiology and Pathology: A Revisit. *Int J Mol Sci* **20**, doi:10.3390/ijms20246110 (2019).
- 6 Starruss, J., de Back, W., Bruschi, L. & Deutsch, A. Morpheus: a user-friendly modeling environment for multiscale and multicellular systems biology. *Bioinformatics* **30**, 1331-1332, doi:10.1093/bioinformatics/btt772 (2014).
- 7 Alamoudi, E. *et al.* FitMultiCell: simulating and parameterizing computational models of multi-scale and multi-cellular processes. *Bioinformatics* **39**, doi:10.1093/bioinformatics/btad674 (2023).

- 8 Kulkarni, A. A. *et al.* An In Vivo Zebrafish Model for Interrogating ROS-Mediated Pancreatic beta-Cell Injury, Response, and Prevention. *Oxid Med Cell Longev* **2018**, 1324739, doi:10.1155/2018/1324739 (2018).
- 9 Geravandi, S., Richardson, S., Pugliese, A. & Maedler, K. Localization of enteroviral RNA within the pancreas in donors with T1D and T1D-associated autoantibodies. *Cell Rep Med* **2**, 100371, doi:10.1016/j.xcrm.2021.100371 (2021).

Dear Dr Ninov,

Thank you for submitting your revised manuscript (EMBOJ-2024-117149R) to The EMBO Journal, as well for your patience with our response. Your amended study was sent back to the three referees for their scientific re-evaluation, and we have received detailed comments from all of them, which I enclose below. As you will see, the experts state that the work has been substantially improved by the revisions and they are now broadly in favour of publication, pending minor revision.

Thus, we are pleased to inform you that your manuscript has been accepted in principle for publication in The EMBO Journal.

Please consider the remaining issues of referees #1 and #3 carefully and amend the manuscript accordingly by complementary data presentation and revisiting the discussion of the results.

We also now need you to take care of a number of issues related to formatting and data presentation as detailed below, which should be addressed at re-submission.

Please contact me at any time if you have additional questions related to below points.

Thank you for giving us the chance to consider your manuscript for The EMBO Journal. I look forward to your final revision.

Again, please contact me at any time if you need any help or have further questions.

Best regards,

Daniel Klimmeck

>> Authors: please clarify the name discrepancy: L. D. S. in our system vs L. F. S. D. in the manuscript.

>> Limit the number of keywords for your study to maximally five.

>> Author Contributions: Please remove the author contributions information from the manuscript text. Note that CRediT has replaced the traditional author contributions section as of now because it offers a systematic machine-readable author contributions format that allows for more effective research assessment. and use the free text boxes beneath each contributing author's name to add specific details on the author's contribution.

More information is available in our guide to authors.
<https://www.embopress.org/page/journal/14602075/authorguide>

>> Rename the current 'Competing Interests' section to 'Disclosure and Competing Interests Statement'.

>> Correct order of manuscript sections: Abstract / Keywords / Introduction / Results / Discussion / Methods / Data Availability / Acknowledgements / Disclosure and competing interests statement // References / Figure legends / Tables and their legends / Expanded View Figure legends

>> References: adjust reference format to EMBO Journal format, 10 authors et al.

>> Figures should be uploaded as individual, high resolution figure files.

>> Appendix file: Supplementary figures should be renamed "Appendix Figure S1" etc. The legends for the suppl. tables should be removed from the appendix file and added directly to the corresponding excel tables in a separate tab/worksheet. The correct nomenclature is "Dataset EV1" etc. The movie legends should be removed from the appendix and zipped to the corresponding movie file. The correct nomenclature is "Movie EV1" etc. The appendix file should be renamed from "Supplementary Data" to "Appendix, and the table of contents should have page numbers added.

>> The PDF uploaded with the title "Mathematical order of beta cell turnover" should be added to the appendix as "Appendix Supplementary Materials"?

>> Add a Reagents and Tools table to the Methods section, listing key reagents, experimental models, software and relevant equipment.

>> Data availability section: Remove the referee token and make sure the dataset privacy is released.

>> Provide a complete set of source data for the study according to the separate instructions by my colleague Hannah Sonntag sent via e-mail.

>> Consider additional changes and comments from our production team as indicated below:

- Data availability section:

1. Please note that the specific URL for GSE261729 dataset is not provided in the data availability statement.
2. Please note that reviewer access code for GSE261729 dataset is not provided in the data availability statement.

- Figure legends:

1. Please note that the legend for figure 5f is mislabelled as figure 5g in the manuscript. This needs to be rectified.
2. Please note that information related to n is missing in the legends of figures 2a', b', c'.
3. Please note that the error bars are not defined in the legends of figures 2a', b', c'; 3b, d; 4f-g, i; 6b-c; 7b.
4. Please note that the white arrows are not defined in the legend of figure 3a, c'. This needs to be rectified.

Referee #1:

The authors have addressed my concerns about the original version of the MS. However, I have a minor comment on their additional results and discussion.

Minor critic

Concerning their statement on the revised version of Supplementary Figure 8B, page 15, line 325, '... suggesting a bystander effect of inflammation', the reviewer does not entirely agree with this statement. As the authors showed in their manuscript, inflammation caused the dedifferentiation of beta cells; the insulin-negative cells might be the dedifferentiated cells and, therefore, not the bystander effect, as the authors stated. The authors should change the statement or perform an experiment to support their hypothesis.

Referee #2:

In the revised version of this manuscript the authors have satisfactorily responded to all the points raised by this referee. The new set of experiments included has greatly improved the work.

Referee #3:

The manuscript by Akhtar and colleagues is significantly improved, though a few issues remain that should be addressed before publication:

- The authors now present data showing that p35 does not completely block apoptosis in the ins:TRPV/csn model. This raises important questions regarding their mathematical models: for example, do the models underestimate the level of apoptosis during the remodeling process? As stated on page 7, the death rate is "set to zero for the p35 genotype" in the model. Alternatively, could the endogenous developmental mechanism triggering apoptosis be more susceptible to p35 protection than the beta cells in the TRPV model? The observed lack of pro-inflammatory macrophages and the significant reduction in overall mpeg+ macrophages in six-month-old p35 fish might support this interpretation.
- The slingshot analysis suggests a progression of stressed beta cells through distinct states identified in the UMAP clusters (Figure 5E), following the trajectory 2-1-3-4-5. However, the model in Figure 5F appears to contradict this, depicting a bifurcation of fates downstream of the dysfunctional islet (perhaps at cluster 3), resulting in either apoptosis (cluster 4) or dedifferentiation (cluster 5). Could there be a resolution between these paradigms where cells initially committed to apoptosis (cluster 4) revert to dedifferentiation (cluster 5), or is this outcome unlikely? Clarifying which model the authors support would strengthen the manuscript. Additionally, the authors should discuss whether the cells in cluster 5 evade stress primarily by downregulating TRPV, as they appear to do with insulin.

Minor points:

- The EdU data should be presented as a percentage of nuclei, similar to PCNA, to allow direct comparison. The current format is inconsistent with other proliferation and apoptosis quantifications in the manuscript and does not easily align with the models.
- The newly included cell count data at 20-27 def should be integrated into the original Figure 1E for clarity and consistency.

Referee #1:

The authors have addressed my concerns about the original version of the MS. However, I have a minor comment on their additional results and discussion.

Minor critic

Concerning their statement on the revised version of Supplementary Figure 8B, page 15, line 325, '... suggesting a bystander effect of inflammation', the reviewer does not entirely agree with this statement. As the authors showed in their manuscript, inflammation caused the dedifferentiation of beta cells; the insulin-negative cells might be the dedifferentiated cells and, therefore, not the bystander effect, as the authors stated. The authors should change the statement or perform an experiment to support their hypothesis.

We thank the reviewer for the positive response to our revisions. The reviewer makes a good point. We have removed this statement from the text.

Referee #2:

In the revised version of this manuscript the authors have satisfactorily responded to all the points raised by this referee.

The new set of experiments included has greatly improved the work.

We thank the reviewer for the positive response to our revisions

Referee #3:

The manuscript by Akhtar and colleagues is significantly improved, though a few issues remain that should be addressed before publication:

We thank the reviewer for the positive response to our revisions. We did our best to address the remaining issues.

- The authors now present data showing that p35 does not completely block apoptosis in the ins:TRPV/csn model. This raises important questions regarding their mathematical models: for example, do the models underestimate the level of apoptosis during the remodeling process? As stated on page 7, the death rate is "set to zero for the p35 genotype" in the model. Alternatively, could the endogenous developmental mechanism triggering apoptosis be more susceptible to p35 protection than the beta cells in the TRPV model? The observed lack of pro-inflammatory macrophages and the significant reduction in overall mpeg+ macrophages in six-month-old p35 fish might support this interpretation.

We agree with the reviewer's view too. Indeed, the strong reduction of macrophages suggests that there are no escaper beta-cells that evade p35. It is possible that the stress on the cells is higher in the TRPV model than during development as the fish also become hyperglycemic. However, it is also possible that the process of dedifferentiation is not blocked by p35, causing a partial rescue. Whether beta cell dedifferentiation also happens during development as it does in the TRPV model and contributes to beta-cell loss is something we would like to follow-up in the future using lineage tracing. We discussed this consideration in the discussion (lines 557-561).

- The slingshot analysis suggests a progression of stressed beta cells through distinct states identified in the UMAP clusters (Figure 5E), following the trajectory 2-1-3-4-5. However, the model in Figure 5F appears to contradict this, depicting a bifurcation of fates downstream of the dysfunctional islet (perhaps at cluster 3), resulting in either apoptosis (cluster 4) or dedifferentiation (cluster 5). Could there be a resolution between these paradigms where cells initially committed to apoptosis (cluster 4) revert to dedifferentiation (cluster 5), or is this outcome unlikely? Clarifying which model the authors support would strengthen the manuscript. Additionally, the authors should discuss whether the cells in cluster 5 evade stress primarily by downregulating TRPV, as they appear to do with insulin.

A common feature of clusters 4 and 5 is the reduced expression of *pdx1* and *insulin*, with cluster 5 showing a more pronounced downregulation of beta-cell markers together with the additional expression of *aldh1a3*. We interpret that under stress, there is a common initial response to downregulate beta-cell markers, however, some beta-cells do not proceed to full dedifferentiation and succumb to the stress, expressing cell death genes such as *casp9* (cluster 4), while others go further down the dedifferentiation path, expressing *aldh1a3* and evading cell death (cluster 5). The slingshot analysis likely captures the signature of this common initial response, subsequently diverging into cluster 5 as the cells bypass the cell death response and proceed towards further dedifferentiation.

Dedifferentiation can alleviate the stress on the beta-cells and help them survive. In this way, our model is correct, stating that beta-cells that receive a stressful signal can either dedifferentiate or undergo cell death as their ultimate fate. Following the reviewer's suggestion, we have now explained this concept in the figure legend pertaining to the schematic of Fig 5F. Since we do not see the expression of cell death genes in cluster 5, we do not think that this subset of cells had initially committed to apoptosis, although we could not fully rule this out and would be interested to test this interesting idea in the future.

TRPV expression was not mapped in the RNA seq as it was not part of the zebrafish genome so we could not check its levels of expression. The reviewer makes a valid point and we have now mentioned it in the discussion (lines 557-559). It is important to note that after TRPV induction, the fish become hyperglycemic and hence, all beta-cells will be equally exposed to this stressor.

Minor points:

- The EdU data should be presented as a percentage of nuclei, similar to PCNA, to allow direct comparison. The current format is inconsistent with other proliferation and apoptosis quantifications in the manuscript and does not easily align with the models.

The EdU data was already presented as a percentage of beta-cell, similar to the PCNA, however the figure was unclearly labeled. We apologize for the confusion. We corrected the graph to indicate Ins+ EdU+ cells (%) as for the PCNA graph. The proliferation is consistent with the PCNA data.

- The newly included cell count data at 20-27 def should be integrated into the original Figure 1E for clarity and consistency.

We integrated the new data as suggested.

Additional correction:

We noticed that the data from human donors provided by our collaborators were initially analyzed using T-test, whereas a one-way ANOVA would have been more appropriate due to having three comparison groups. We rerun the analysis using ANOVA, which confirmed a significant increase in macrophages in the islet of donors with Type 1 diabetes (Fig. EV4). This supports our model that macrophages recruit to the islets in the course of beta cell death in humans. The data from the T1D cohort makes a strong point that beta-cell death is linked to macrophage colonization. While showing a tendency for more macrophages, the AAB+ group did not reach significance. Hence, we toned-down the conclusion that macrophages are also increased earlier at the AAB+ stage. We prefer to leave this question open for subsequent analysis when a larger samples size would become available.

Dear Dr. Ninov,

Thank you for submitting the revised version of your manuscript. I have now evaluated your amended manuscript and concluded that the remaining minor concerns have been sufficiently addressed.

I am thus pleased to inform you that your manuscript has been accepted for publication in the EMBO Journal.

Related I would like to hereby ask your consent on keeping the referee figures included in this file.

On a different note, I would like to alert you that EMBO Press offers a format for a video-synopsis of work published with us, which essentially is a short, author-generated film explaining the core findings in hand drawings, and, as we believe, can be very useful to increase visibility of the work. Please see the following link for representative examples and their integration into the article web page:

<https://www.embopress.org/doi/full/10.15252/emj.2019103932>

Best regards,

Daniel Klimmeck

Daniel Klimmeck, PhD
Senior Editor
The EMBO Journal
EMBO
Postfach 1022-40
Meyerohofstrasse 1
D-69117 Heidelberg
contact@embojournal.org
Submit at: <http://emboj.msubmit.net>
